# Initial insights from a global database of rainfall-induced landslide inventories: the weak influence of slope and strong influence of total storm rainfall

Odin Marc[1], André Stumpf[1], Jean-Philippe Malet[1], Marielle Gosset[2], Taro Uchida[3], and Shou-Hao Chiang[4]

[1]École et Observatoire des Sciences de la Terre − Institut de Physique du Globe de Strasbourg, Centre National de la Recherche Scientifique UMR 7516, University of Strasbourg, 67084 Strasbourg Cedex, France
[2]Géoscience Environnement Toulouse,Toulouse,France
[3]National Institute for Land and Infrastructure Management, Research Center for Disaster Risk Management, Tsukuba, Japan
[4]Center for Space and Remote Sensing Research, National Central University, Taoyuan City 32001, Taiwan

**Correspondence:** Odin Marc (odin.marc@unistra.fr)

**Abstract.** Rainfall-induced landslides are a common and significant source of damage and fatality worldwide. Still, we have very little understanding of the quantity and properties of landsliding that can be expected for a given storm and a given landscape, mostly because we have very few inventories of rainfall-induced landslides caused by single storms. Here we present six new comprehensive landslide event inventories coincident with well identified rainfall events. Combining these datasets, with two previously published datasets, we study their statistical properties and their relations to topographic slope distribution and storm properties. Landslide metrics (such as total landsliding, peak landslide density or landslide distribution area) vary across 2 to 3 orders of magnitude but strongly correlate with the storm total rainfall, varying over almost 2 orders of magnitude for these events. Applying a normalization on the landslide runout distances increases these correlations and also reveals a positive influence of total rainfall on the proportion of large landslides. The nonlinear scaling of landslide density with total rainfall should be further constrained with additional cases and incorporation of landscape properties such as regolith depth, typical strength or permeability estimates. We also observe that, rainfall-induced landslides do not occur preferentially on the steepest slopes of the landscape, contrary to observations from earthquake-induced landslides. This may due to the preferential failures of larger drainage area patches with intermediate slopes or to the lower pore-water pressure accumulation in fast draining steep slopes. The database could be used for further comparison with spatially resolved rainfall estimates and with empirical or mechanistic landslide event modeling.

## 1 Introduction

Landslides associated with heavy rainfall cause significant economic losses and may injure several thousand people a year worldwide (Petley, 2012). In addition, the frequency of landsliding increases with that of extreme rainfall events (Kirschbaum et al., 2012), which are expected to increase with global change (Gariano and Guzzetti, 2016). Landslides are also recognized as a major geomorphic agent contributing to erosion and sediment yield in mountainous terrain (Hovius et al., 1997; Blodgett and Isacks, 2007). Yet, constraining quantitatively relationships between landslides and rainfall metrics remains difficult.

We have some theoretical understanding of how rainfall, through water infiltration in the ground, can increase pore-water pressures and trigger failures (Van Asch et al., 1999; Iverson, 2000). Therefore a variety of mechanistic models have been developed, usually by coupling a shallow hydrological model to a slope failure criterium (e.g., Montgomery and Dietrich, 1994; Baum et al., 2010; Arnone et al., 2011; Lehmann and Or, 2012; von Ruette et al., 2013). However, such deterministic approaches require not only appropriate physical laws, but also an accurate and fine-scale quantification of many input parameters (topography, cohesion, permeability, rainfall pattern)(Uchida et al., 2011). In most places, such level of detailed information is currently unavailable, rendering deterministic approaches hardly applicable.

Data-driven studies have mostly focused on using precise information on individual landslide location and timing to decipher thresholds, typically based on preceding rainfall intensity and duration, at which landslide would initiate (Caine, 1980; Guzzetti et al., 2008, and references therein). Although useful for hazard and early-warning purposes (e.g., Keefer et al., 1987), such approaches completely ignore the number, size and other properties of landslides that can be triggered by a rainfall event, while these properties are useful, if not required for most natural hazard or geophysical applications. In order to understand the importance of rainfall on erosion rates or to anticipate landslide hazard associated with emerging cyclones and heavy rainstorms, it is highly desirable to quantitatively relate the properties of a landslide event $\mathbf{L}$ (total area, volume, size distribution) to the combination of site susceptibility, $\mathbf{s}$, and rainfall forcing, $\mathbf{f}$, properties, or equivalently to develop scaling relations of the form of:

$$\mathbf{L} = g(\mathbf{s}(slope,\ soil\ thickness,\ strength,\ permeability, ...), \mathbf{f}(total\ rainfall,\ intensity,\ antecedent\ rainfall, ...)) \quad (1)$$

Note that variables in such equation may be a statistical description at the catchment or landscape scale (being a simple mean or other moments of the distribution), and thus may not describe the finescale variability required by mechanistic models. Although being simplified versions of mechanistic models, such scaling laws can be useful to describe average properties of the phenomena, here a population of landslides associated with a constrained trigger. The value of statistical or semi-deterministic approaches is that they are able to predict accurately global properties, while circumventing the difficulties of predicting specific local properties of individual landslides. Indeed, such scaling laws would allow prediction in data scarce regions and possibly at various scales (hillslope scale, catchment scale, region scale, etc). This approach has driven important progress for both the understanding and hazard management of earthquake-induced landslides, thanks to the introduction of purely empirical, physically inspired or mixed functional relations in the form of Eq. 1 (e.g., Jibson et al., 2000; Meunier et al., 2007, 2013; Nowicki et al., 2014; Marc et al., 2016, 2017). This progress has been possible thanks to, first detailed investigation of individual case studies with comprehensive landslide event inventories (e.g., Harp and Jibson, 1996; Liao and Lee, 2000; Yagi

et al., 2009) and next through their combined analysis as aggregated databases (Marc et al., 2016, 2017; Tanyaş et al., 2017). By comprehensive event-inventories we mean that all landslides larger than a given size were mapped, and that the spatial extent of the imagery allowed to observe the landslide density fading away in all direction, tracking the reduction of the forcing intensity of the triggering event, whether shaking or rainfall.

In contrast, very few studies on rainfall induced landslides are based on comprehensive, event-inventories. Some studies are based on individual landslide information. For example, Saito et al. (2014) studied 4744 landslides in Japan, that occurred between 2001 and 2011, to better understand which rainfall property controls landslide size. This dataset, aggregating small subset of the landslides triggered by rainfall events, misses the vast majority of landslides (for example, in Japan, Typhoon Talas alone caused a similar amount of landslides in a few days). It is therefore insufficient for more advanced statistical analy-

ses. At global scale, Kirschbaum et al. (2009), presented a catalog containing information on 1130 landslide events worldwide, occurring in 2003, 2007 and 2008. With this catalog they underlined the correlation between extreme rainfall and landslid-ing (Kirschbaum et al., 2012). However, such catalogs, mainly based on reports from various kinds, gather very fragmentary knowledge, and contain little quantitative information on landsliding. Thus, we consider that neither studies based on small sample of individual landslide or on global-scale analysis will be able to constrain effectively Eq 1, and that storm scale de-

tailed information is needed.

Although relatively rare, some case studies based on fragmentary event inventories exist (and are briefly reviewed in the next section) but they may contain too few landslides for statistical analyses or may be biased to specific locations (e.g., along roads or near settlements, within weak lithological units, near rivers), thus complicating the deconvolution of forcing and site influences. However, in theory, satellite imagery allows comprehensive mapping of landslides larger than the resolution limit,

across all catchments affected by a large storm. In practice, obtaining useful images strictly constraining the landsliding caused by a single storm is not always possible, mainly because of cloud coverage, and detailed mapping across vast areas represent a significant work effort. As a result, landslide inventories triggered by rainfall during a whole season or a few years are used for testing mechanistic models (e.g., Baum et al., 2010; Arnone et al., 2011).

The purpose for this work is to present a compilation of new and past comprehensive rainfall-induced landslide (RIL) inven-

tories, each containing the landslide population associated with an identified storm. They constitute the core of an expandable database, essential for further research. We first briefly review existing comprehensive and partially complete inventory asso-ciated with specific storms. Then we present six new inventories and analyze their statistical properties in terms of size (total area, landslide density), geometry (length, width and depth) and relation to topographic slopes. We further analyze and discuss the properties relative to rainfall observations in those cases and conclude on the various insights that can be derived from such

inventory compilation.

## 2   Data and methods

### 2.1   Review of pre-existing datasets

An in-depth literature review revealed that very few comprehensive, digital, RIL inventories have been published, such as the Colorado 1999 and Micronesia 2002 events detailed below. If we look for partial inventories, in which landslides have been
mapped comprehensively in limited zones affected by a storm, a few more datasets exists.

For example, hurricane Mitch hit central America at the end of 1998 and triggered thousands of landslides across several countries. The rainfall was record breaking in many places, with rain gauges recording up to 900, 1100 and 1500 mm in Honduras, Guatemala and Nicaragua (Bucknam et al., 2001; Cannon et al., 2001; Crone et al., 2001; Harp et al., 2002). In the following weeks, the USGS performed an number of air-surveys, identified the most affected areas in these three countries as well as in
El Salvador (where the rainfall amount was less), and mapped a large number of the failures based on aerial photographs. The resolution of the optical imagery allowed to distinguish failures down to a relatively small size ($<100$ m$^2$), but the mapping amalgamated multiple failures into single polygons, and combined very long debris flow paths and/or channel deposits to the source areas. Last, the mapping was only performed in a number of isolated areas, that were "the most affected", and it is unclear how much landsliding was present in the unmapped zones. Because of these limitations, we did not investigate this
case in detail here but note that these inventories may be corrected and used by later studies. Similarly in a number of studies, inventory of all the landslides caused by a given storm in a specific catchment or geographic zone can be found, in Liguria 2000 (Guzzetti et al., 2004), Umbria 2004 (Cardinali et al., 2006), Sicily 2009 (Ardizzone et al., 2012), Peru 2010 (Clark et al., 2016), Thailand 2011 (Ono et al., 2014) and Myanmar 2015 (Mondini, 2017) and in Taiwan for ten typhoons between 2001 and 2009 (Chen et al., 2013). These inventories could not constrain the total landslide response to a storm, but may allow to
constrain relationships between landslide properties and local rainfall properties, provided that enough landslide have been mapped for statistical analysis (e.g., >50-100) and without any systematic sampling bias. However, a detailed assessment of these datasets properties and of their relation to rainfall is out of the scope of this study although it would probably complement interestingly our work in the future.

In this study we analyzed two datasets published previously by the USGS. First, an afternoon rain on the 28th of July 1999 that triggered numerous landslides and debris flows in the Colorado front range (Godt and Coe, 2007). Based on aerial photographs interpretation and field inspection, landslides were mapped as polygons containing source areas, debris flow travel and deposition zones. Initiation point were assumed to be the highest point upslope of each mapped landslides. In 57 out of 328 polygons multiple initiation points (2 to $> 15$) were mapped for multi-headed polygons (Godt and Coe, 2007). These polygons
are among the largest of the inventory and represent 61% of the total landslide area. The surface of the source areas were often of similar width, suggesting equivalent contribution from each source to the transport and deposit areas, and rendering a manual splitting impractical. Thus, we instead conserve multi-headed polygons and we use the whole landslide area, $A_l$, perimeter, $P_l$, and number of source, $N_s$, for each multi-headed polygon to derive an equivalent area and perimeter associated with each source: $A_l^* = A_l/N_s$ and $P_s = P_l/N_s$. This first order approach underestimates the perimeter of each components

by one width (the segment that would be added for each necessary split), however this underestimation decreases with the length/width ratio of the polygons, and is already below 10% for $L/W > 4$. In any case, this assumption does not affect the total area affected, but it changes the landslide frequency-area and frequency-width distributions, and all terms derived from them.

The second dataset contains landslides caused by a summer typhoon in July 2002, mapped exhaustively with aerial photos on the islands of Micronesia (Harp et al., 2004). We digitized the original maps based on strong contrast between red polygons and the rest of the maps. A few artifacts due to this image processing were removed and a few amalgams were split. Again, scarps and deposits are not differentiated.

## 2.2   New comprehensive inventories of rainfall-induced landslides

We present the mapping methodology and imagery (Table Suppl. 1) used to produce six additional inventories. Here we consider landslides as a rapid downslope transport of material, disturbing vegetation outside of the fluvial domain, which we define by visible water flow in the imagery. We also consider individual landslides with a single source or scar areas to avoid amalgamation, and split polygons when necessary. Although the transition between hillslopes and channel may be blurry and in part subjective, the width estimation (cf 2.4) will mitigate variations of the transport length, as long as large alluviated, or flooded

areas are not mapped as landslide deposits. Still the limit between scar, transport and deposit areas could rarely be detected with the available imagery, and all polygons consider the whole disturbed areas on the hillslopes. Subset of the inventories in Taiwan 2009 and Brazil 2011 were produced with automatic algorithm, and then edited and corrected manually, while all others were manually mapped.

In 2008 around the Brazilian town of Blumenau, several days of intense rainfall at the end of a very wet fall triggered

widespread landsliding and flooding, with some partial inventories published in the Brazilian literature (e.g., Pozzobon, 2013; Camargo, 2015) but no analyses in the international literature. The detection and manual mapping of landslides as georeferenced polygons was primarily done with a pair of Landsat 5 cloud free images (02/01/2009 and 03/02/2008). The coarse resolution (30 m) of the images allowed only to locate vegetation disturbances and accurate landslide delineation was only possible for the largest events. Therefore we used extensively high-resolution imagery available in Google Earth ( over $> 90\%$

of the AOI) acquired in May-June 2009 in most areas, and in 2010-2012 elsewhere, where scars were still visible. To avoid mapping post-event landslides, we mapped only the ones corresponding to vegetation radiometric index (e.g., NDVI) reduction for the pair of Landsat 5 images, present even for sub-pixel landslides (e.g., 10x5m). Thus the landslide mapping could be confirmed for $\sim$90% of the mapped polygons, and man-made digging or deforestation occurring on steep slopes could be avoided. This approach avoid amalgamating groups of neighboring landslides and to map very small landslides ($\sim$1 pixel in Landsat 5

images). However, some detailed field mapping in the surrounding of Blumenau reports up to twice the number of landslides that we observed (Pozzobon, 2013) indicating that we still miss a number of small events. Nevertheless, these landslides must be quite small (not visible in $\sim$ 1 m resolution imagery) and likely do not affect any of our statistics (area, volume, slope) apart the total number of landslides.

The same approach was used to map the intense landsliding caused by a few days of intense rainfall between the $10^{th}$ and the

$12^{th}$ of January 2011 (Netto et al., 2013), in the mountains northeast of Rio de Janeiro. Near Teresopolis, we used first a pan-sharpened (10 m) EO-ALI and a 30 m Landsat 7 images from February 2011 for co-registration and ortho-rectification. $> 95\%$ of the slides were cross-checked in Google Earth based on images from the 20th and 24th o fJanuary 2011 (Fig 1) and where clouds or no images where available we mapped landslides directly from Google Earth (available over $>90\%$ of the AOI), even if poor ortho-rectification may create geometric distortions. Closer to Nova Friburgo, we used a pair of very high resolution Geoeye-1 image (2/0.5 m resolution in multispectral/panchromatic) from the 26th of May 2010 and the 20th of January 2011. On these images we applied the methods presented by Stumpf et al. (2014), to classify the whole image, detecting $> 90\%$ of the landslides we could observe manually, but also including false positive. Thus, we screened manually the image to remove agricultural field, inundated areas, channel deposits that were included and split the amalgamated landslides, very frequent given the important clusters of landslides in many part of the image. This correction seems sufficient given that landslide size distribution for the three subparts of the inventory are consistent (Fig. Suppl. 1). Polygons from the automatic classification display a slightly larger equivalent length/width ratio, maybe because some amalgamated polygons have been missed or simply because the classification allows hollow polygons, biasing upward the length/width estimate based on a perimeter/area ratio (cf 2.4).

In the first days of September 2011 (1st-4th), typhoon Talas triggered very heavy rainfall on the Kii Peninsula, in Japan, resulting in several thousands of landslides. For disaster emergency response, the National Institute for Land and Infrastructure Management of Japan mapped landslides across most of the affected areas based mainly on post-typhoon aerial photographs and occasionally on Google Earth imagery (Uchida et al., 2012). Screening antecedent imagery (2010-2011) from Google Earth and Landsat 5 we identified and removed a few hundreds pre-Talas polygons, mostly within 5 km of 136,25°E/34,29°N and 135,9°E/34,20°. With Google Earth we could validate NILIM mapping over about 85% of the AOI and we added almost 200 polygons in areas were aerial photographs were not taken, and split many large or multi-headed polygons that were amalgamated. Some polygons had distorted geometry or exaggerated width, most likely due to poor ortho-rectification of the aerial imagery and/or time-constrains for the mapping. We could not systematically check all polygons, but we checked and corrected all polygons larger than $30,000 m^2$ (3% of the catalog but representing 45% of the total area). We consider that the remaining distortions for some of the smaller polygons have minor impacts on the statistics discussed in the next sections.

In Taiwan, we collected landslide datasets associated with the 2008 Kalmaegi (16-18th Jul.) and 2009 Morakot (6-10th Aug.) typhoons, partially described by Chen et al. (2013). For 2008, we compared multispectral composite images and NDVI changes between (30 m) Landsat 5 images taken on the 06/21, 07/07 and 07/23. The image from 07/07 is covered by clouds and light fog in many parts but allows identifying that most places affected by landslides in the last images were still vegetated at this time. Thus all new landslides are attributed to the rainfall from typhoon Kalmeagi. For 2009, landslides were mapped with pre- and post-event FORMOSAT-2 satellite images (2 m panchromatic and 8 m multi-spectral) (Chang et al., 2014). To cover most of the island, we mosaicked multiple mostly cloud-free pre-event (01/14, 05/08, 05/09, 05/10, 06/06 of 2009) and post-event (08/17, 08/19, 08/21, 08/28, 08/30, 09/06 of 2009) images. For subsets of the inventory, especially to the East of the main divide, landslides were significantly amalgamated and bundled with river channel alluviation. We thus manually split the polygons and removed the channel areas. In a few areas with clouds ($< 5\%$ of the AOI) in the post-event mosaic, we mapped

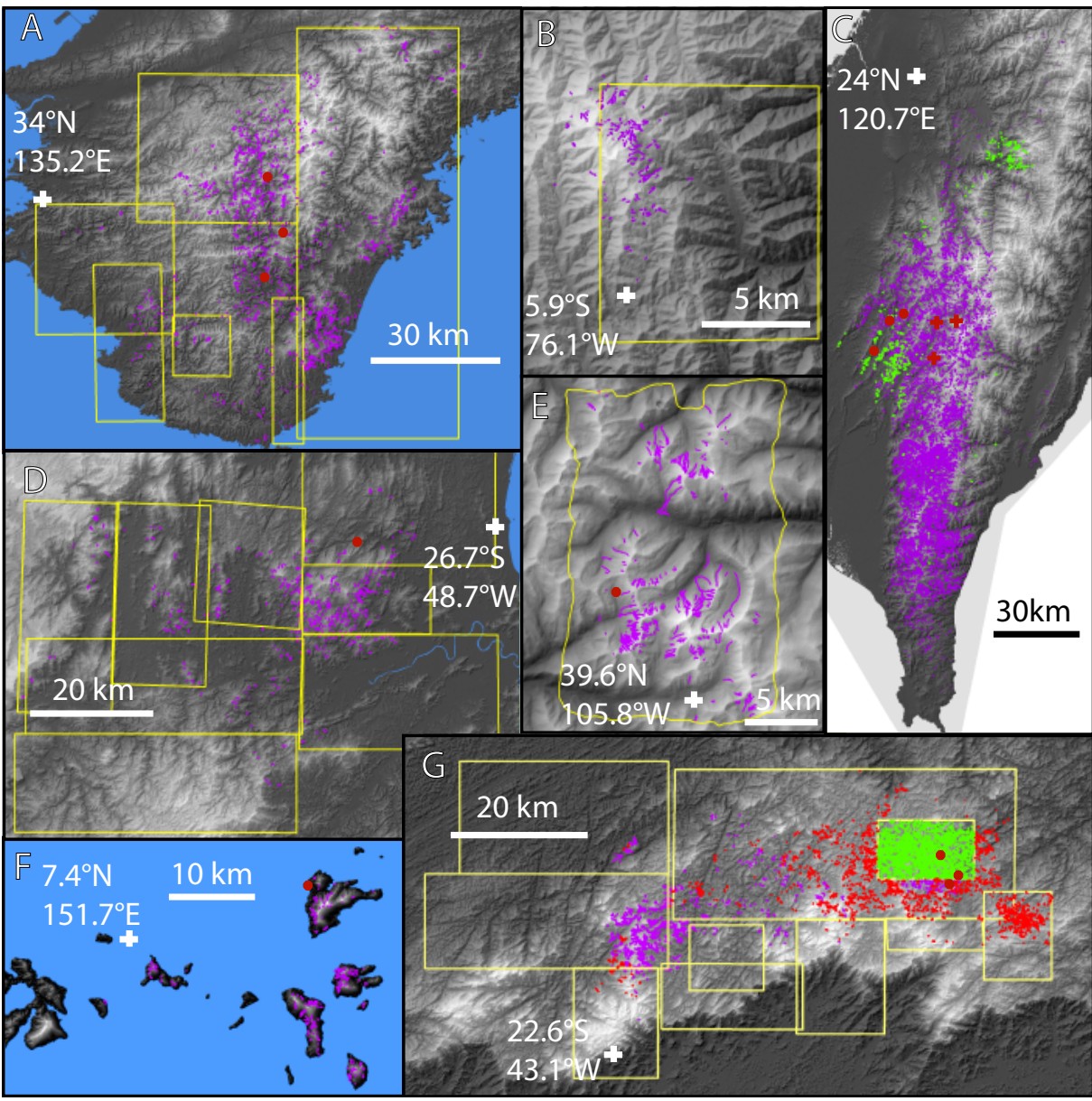

**Figure 1.** Landslides inventory superimposed on digital surface model for the events in Japan 2011 (A), Brazil 2008 (B), Micronesia 2002 (C), Colombia 2015 (D), Colorado 2002 (E), Brazil 2011 (F) and Taiwan 2008 and 2009 (G). Landslides are in purple, rain gauges used in this study are in red dots (and red crosses for Taiwan 2009), and the yellow frames show the availability of high resolution imagery (Google Earth) used to check or perform the mapping. In G, the pink dots are landslides from 2008, while purple dots are from 2009. In F, purple, pink and cyan are landslides mapped from EO-ALI, Google Earth and automatic classification of Geoeye image, respectively.

with Landsat 5 images (from 24th of June and 12th of September 2009), even if the spatial resolution limit may have censored the smallest landslides in these zones. Special attention was given to the separation of individual landslides by systematically checking and splitting polygons above 0.1 km$^2$ (2% of the catalog but representing 30% and 60% of the total area and volume). However, it is clear that a number of smaller slides are missed or merged with large ones, and therefore although total land-

sliding and landslide locations on slope may be well represented, the size distribution of this catalog must be biased to some extent.

Between 15 and 17th of May 2015, heavy rainfall in the mountains above the village of Salgar, Colombia, triggered catastrophic landslides and debris flow (>80 deaths). Landslide mapping was carried out by comparing a (10 m) Sentinel 2 image from the 21st of July 2016 and a pan-sharpened (15 m) Landsat 8 image from the 19th of July and 26th December of 2014.

These images were selected for their absence of clouds, good conditions of light and similarity. High spatial resolution imagery from Google Earth, dated from the 31st of May 2015 shows fresh scars consistent with our mapping over most of the area (Fig 1), and we assumed that the remaining landslides ($< 15\%$ of the inventory) were also triggered by the same rainfall event.

## 2.3   Rainfall data

Rainfall data quality and amount are very variable for the different events, from 0 or 1 single gauge (For Micronesia or Colom-

bia), to dense gauge network and potentially weather radar coverage in Japan, Taiwan and Brazil. Therefore we selected a simple index that could be obtained for each case in order to discuss potential rainfall controls on the landslide properties. For each case we calculated an estimate of total rainfall, $R_t$, duration, $D$, and a peak rainfall intensity over 3 hours, $I3$ (Table 1). Note that these variables do not represent an average value within the whole footprint of the storm, but rather a maximal forcing, usually colocated with the areas where landsliding was the most intense (Fig 1) and derived mostly from one or a few

rain-gauges. Thus these indexes may be taken as a storm magnitude. A more detailed analysis of the spatio-temporal pattern of the rainfall and of its relations to the spatial pattern of landsliding is highly desirable, but challenging and is left for a future study.

The estimates from Taiwan and Japan are based on hourly gauge measurements from the Japan Meteorological Agency and Taiwan Institute for Flood and Typhoon research. In each case we took the three closest gauges within 5 to 15 km from the

areas with the highest landslide densities (in 0.05 by 0.05° window) (Fig Suppl. 2) and computed their average properties (Fig 2). Minimum and maximum single gage measurements give a coarse measure of the uncertainty. A single gauge is available in Micronesia and we used the hourly rainfall from 1st to 3rd of July 2002 reported in Harp et al. (2004). For Colorado, we used the hourly rainfall from the rain gauge at Grizzly peak, closest to intense landsliding and reported by Godt and Coe (2007). For this event radar data indicates very localized high intensity precipitation located on the peaks where the debris flows occurred

(Godt and Coe, 2007)) and suggests that the single closest gauge is more representative than averaging with the other nearby ones. For the event in Brazil 2008 we considered the total daily rainfall from Luis Alves station (Fig 1), where more than 130 mm per day were accumulated on the $21^{st}$, $22^{nd}$ and $23^{rd}$ of November and 250 mm on the $25^{th}$, and intensity going up to 50 mm/hr (Camargo, 2015). These days were also preceded by abnormally wet period, with November 2008 accumulating $\sim$ 1000 mm, 7 times the long term average for this month. In 2011 in Brazil, hourly rain data at Sitio Sao Paulista reports 200

**Table 1.** Rainfall data summary, containing the total rainfall, duration and maximum 3-hours intensity for each storms. For TW8, TW9 and J11, we indicate the range for the three indexes that could be estimated from three gauges near the zone of maximal landsliding. We cannot perform this analysis for MI2 and C99, and can only assess a range of $R_t$ for B08 and B11. For C15, we indicate by a star that we could only access satellite based rainfall estimates (GSMaP version 7 ungauged products, see Fig Suppl. 3). Reference are as follows, 1: (Godt and Coe, 2007), 2: (Harp et al., 2004), 3:(Camargo, 2015) , 4:(Netto et al., 2013) .

| Event | C99 | MI2 | B08 | TW8 | TW9 | B11 | J11 | C15 |
|---|---|---|---|---|---|---|---|---|
| $R_t$, mm | 45 | 500 | 695[680-800?] | 670[600-740] | 2500[2100-2800] | 280[200-320] | 1300[1000-1500] | 65[10-75]* |
| $D$, hours | 4 | 20 | 100 | 24 | 105 | 36 | 62 | 10* |
| $I3$, mm.hr$^{-1}$ | 13 | 65 | 30 | 92[78-116] | 85[83-87] | 55 | 58[38-88] | 8* |
| Ref. | 1 | 2 | 3 | Us | Us | 4 | Us | GSMaP |

mm in 8 hours before gauge failure, while there and at nearby sites, the cumulative rainfall was $\sim 280$ mm from the 10th to the morning of the 12th January (Netto et al., 2013). For these cases, raingages give a trustworthy estimate of the local rainfall, but are not constraining the large scale rainfall pattern. Last, in Colombia, we could not find data from any nearby rain-gauge and we thus use rainfall estimates from the GSMaP global satellite products (Kubota et al., 2006; Ushio et al., 2009) (Fig Suppl.

3). Here, the minimum, mean and maximum rainfall are obtained by considering the triggering storm as the raining period at the time of debris flow occurrence, or the one from the previous day or merging both events, respectively (Fig Suppl. 3).

Defining storm duration accurately requires defining thresholds on rainfall intensity over given periods, to delimit the storm start and end. Given the variable quality of our data we limit ourself to a first order estimate of the continuous period when rainfall was sustained (i.e., $I3 > 3$ mm.hr$^{-1}$). We consider these durations accurate within 10-20% for the events with overall

hourly data. For the less constrained cases B08, B11 and C15 duration is more uncertain. In any case, for these 8 storms, we note a strong correlation between $D$ and $R_t$ and $I3$ and $R_t$ (for power-law scalings, $R^2 = 0.9$ and $R^2 = 0.8$, respectively (Fig Suppl 4). Thus, given that spatial and temporal length scales are often linked in meteorology, the long events causing larger rainfall may also have larger footprints.

**2.4 Landslide area, width and volume**

Landslide planview area and perimeter are directly obtained from each polygon. However, these values represent the total area disturbed, that is the scar, deposit and runout areas, because a systematic delineation of the scar was not possible from most of the imagery. This means that landslide size statistics are resulting from processes affecting both landslide triggering and runout. Landslide volume, estimated based on area, may also be overestimated for long runout slides. Therefore we propose

here a simple way to normalize for landslide runout and obtain an estimate of the scar area.

Following Marc and Hovius (2015), we computed an equivalent ellipse aspect ratio, $K$, using the area and perimeter of each polygons. For polygon with simple geometries, $K$ is close to the actual length/width ratio, but this is a measure that also increases with polygon roughness or branching, and therefore with amalgamation (Marc and Hovius, 2015). Assuming an elliptic

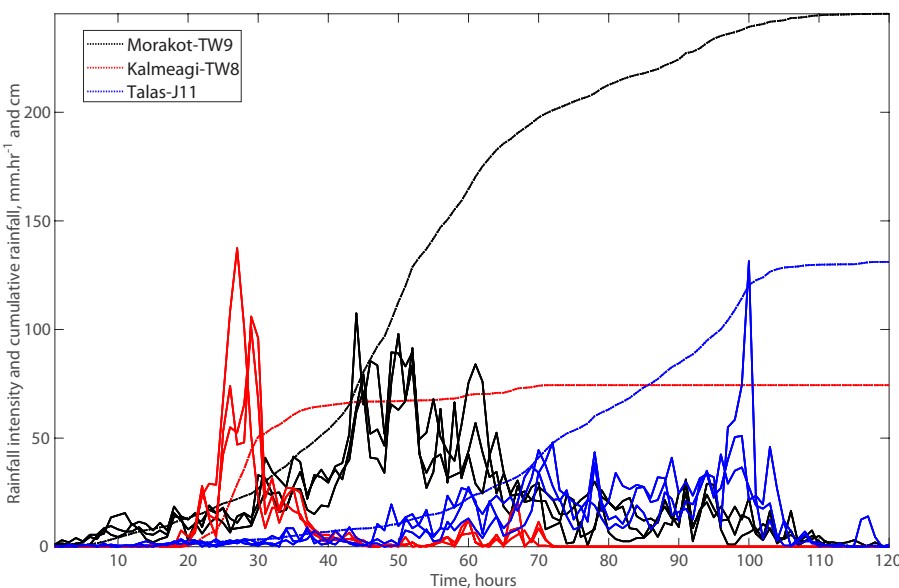

**Figure 2.** Rainfall history for typhoons Kalmaegi (TW8), Morakot (TW9), Talas (J11). For each event, hourly intensity is shown with solid curves for three gauges nearby the area with most intense landsliding (see Fig 1 for locations). Dashed lines represent the mean cumulative rainfall from the three gauges.

shape, polygon area can be approximated by $\pi LW/4$ with $L$ and $W$ being the polygon full length and width, respectively. This allows us to estimate $W \simeq \sqrt{4A/\pi K}$. To validate this geometric method to retrieve landslide width, we measured systematically the width of 418 randomly selected landslides across a wide range of polygon areas and aspect ratios, belonging to four inventories: J11, TW8, B11, and C15. For each polygon, we focused on the upper part of the landslide only, the likely scar, and

5   averaged 4 width (i.e., length perpendicular to flow) measurements made in arcGIS. The width estimated based on P and A are within 30% and 50% of the measured width for 72% and 92% of the polygons, respectively (Fig Suppl 5).We do not observed a trend in bias with area nor aspect ratio, except perhaps for the automatically mapped landslide in B11, where high aspect ratio correlates with underestimated width. Thus, for correctly mapped polygons we can use $P$ and $A$ to derive $W$ and a proxy of landslide scar area, $A_s \sim 1.5W^2$. We assume landslide scars have an aspect ratio of 1.5, as it was found to be the mean

10   aspect ratio found across a wide range of landslide size within a global database of 277 measured landslide geometries (Domej et al., 2017). Even if this equivalent scar area may not exactly correspond to the real landslide scar, it effectively removes the contribution of the landslide runout to the landslide size and allows to compare different size distributions while reducing the impact of variable runout distances.

We also assessed how using $A_s$ affects estimates of landslide volumes and erosion, by computing landslide volume with the

15   total landslide area and with $A_s$ only. In both cases, we used $V = \alpha A^\gamma$, with $\alpha$ and $\gamma$ and their $1\sigma$ globally derived by Larsen et al. (2010). Given that soil and bedrock slides have different shape and that soil slides are rarely larger than $10^5$ m$^2$ ($10^4$ m$^2$ for soil scars) (Larsen et al., 2010), we used the "all landslide" parameters ($\gamma = 1.332 \pm 0.005$; $log_{10}(\alpha) = -0.836 \pm 0.015$) when

$A < 10^5$, and the "bedrock" parameters ($\gamma = 1.35 \pm 0.01$; $log_{10}(\alpha) = -0.73 \pm 0.06$) for larger landslides. Similarly we used the soil scars ($\gamma = 1.262 \pm 0.009$; $log_{10}(\alpha) = -0.649 \pm 0.021$) and bedrock scars ($\gamma = 1.41 \pm 0.02$; $log_{10}(\alpha) = -0.63 \pm 0.06$) for $A_s < 10^4$ m$^2$ and $A_s >= 10^4$ m$^2$, respectively (Larsen et al., 2010). Marc et al. (2016) proposed a rudimentary version of such a runout correction, where they effectively reduced landslide area by a factor 2 for mixed landslides and 3 for bedrock

landslides, noting that volumes derived in this way were closer to field estimates for large landslides than without correction. Uncertainties in this approach include the $1\sigma$ variability of the coefficient and exponent of the landslide area-volume relations given above, and an assumed standard deviation of 20% of the mapped area. These uncertainties were propagated into the volume estimates using a Gaussian distribution. The standard deviation on the total landslide volumes for the whole catalogs or for local subsets, were calculated assuming that the volume of each individual landslide was unrelated to that of any other,

thus, ignoring possible co-variance. Although estimated $2\sigma$ for single landslides is typically from 60 to 100% of the individual volume, the $2\sigma$ for the total volume of the whole catalog is below 10% for the eight datasets. However, for subset with fewer landslides and with volume dominated by large ones, typical when we compute the total landslide volume density in small area (e.g., 0.05°), $2\sigma$ uncertainty reaches 40-60%. We note however, that these uncertainties estimates do not consider potential errors in the identification of landslides, either missed because of occasional shadows or clouds or erroneously attributed to

the storm. Such uncertainty is hard to quantify but must scale with the area obscured in pre- and post-imagery. In most cases multiple pre- and post-event images mean that obscured areas represent typically less than 10% of the affected area, and such errors may be between few to $\sim 20\%$ of the total area or volume, depending on whether obscured areas contain landslide density higher or lower than the average observed throughout the affected area. Last, resolution may not allow to detect the small landslides and in some cases the landslide number may be significantly underestimates, but not the total area and volume

dominated by the larger landslides.

Last, for each inventory, we estimated the landslide distribution area, that is the size of the region within which landslide are distributed. Based on the landslide inventories we could delineate an envelope containing the overall landsliding. As discussed by Marc et al. (2017), such delineation is prone to high uncertainties as it is very dependent on individual isolated landslides. Thus for all cases, we give a range of distribution area, where the upper bound is a convex hull encompassing all the mapped

landslides, while the lower bound is an envelope ignoring isolated and remote landslides (i.e., single or small cluster of landslides without other landslides within 5-10 km), if any. Although the spread can be large in absolute value, both approaches yield the same order of magnitude.

## 3 Results

The inventories contain from $\sim 200$ to $>15,000$ landslide polygons, representing total areas and total volumes (from scars)

from 0.2 to 200 km$^2$ and 0.3 to 1000 Mm$^3$, respectively. The triggering rainfalls are characterized by a total precipitation of $\sim 50$ to 2500 mm in period ranging from 4h to 4 days, and caused landslides within areas ranging from $\sim 50$ to 10,000 km$^2$. Although the dominant landslide types are soil and regolith slumps, a number of large deep-seated bedrock landslides are also present in the inventories associated to the Talas and Morakot typhoons (Saito and Matsuyama, 2012; Chen et al., 2013). A

more detailed description of the landslide types and materials involved was not possible with the available imagery; thus our analysis does not consider landslide types. In the next sections, we present results obtained from these inventories in terms of landslide size statistics, landslide spatial patterns and relation to slope, before correlating these landslide properties to rainfall parameters.

## 3.1 Landslide properties

### 3.1.1 Landslide size statistics

Frequency size distribution of landslide inventories have typically been fit by power-law tailed distributions, above a certain modal size (Hovius et al., 1997; Malamud et al., 2004). The modes and the decay exponents of these distributions are mainly related to the lithology (mechanical strength) or topographic landscape properties (i.e., susceptibility related) (Stark and Guzzetti, 2009; Frattini and Crosta, 2013; Katz et al., 2014; Milledge et al., 2014). Some authors suggested that this behavior could also be affected by the forcing processes. For example analyzing earthquake induced landslide catalogs, it was found that deeper earthquakes, thus with weaker strong-motions, have a smaller proportion of large landslides (Marc et al., 2016). Based on theoretical arguments, it has been proposed that short high intensity rainfall could cause pulses of high pore-water pressures at the soil-bedrock transition, initiating mainly small shallow landslides, while, long duration rainfall with large total precipitation could provoke significant elevation of the water table and trigger large, deep-seated landslides (Van Asch et al., 1999). To our knowledge, only few empirical evidences have supported these assumptions, and we discuss next how our data compare to these ideas.

All landslide size distributions present a roll-over and then a steep decay (Fig 3). The modal landslide area varies between $\sim 3000$ m$^2$ for TW8 and $\sim 300$ m$^2$ for B11, while the largest landslides are $\sim 0.1$ km$^2$ for most events and reach $\sim 0.4$ and 2.8 km$^2$ for J11 and TW9, respectively. The roll-over position certainly relates partly to the spatial resolution and acquisition parameters of the images (Stark and Hovius, 2001)(e.g., for TW8 and B08 where landslides where mostly mapped on coarse spatial resolution image compared to aerial photographs for C99, J11). However, mechanical parameters are also expected to influence the roll-over position (Stark and Guzzetti, 2009; Frattini and Crosta, 2013), as suggested by the fact that MI2, mapped with 1m resolution aerial imagery, has larger modal area than C15 mapped with 10 m Sentinel-2 satellite imagery. Following Malamud et al. (2004) we use maximum likelihood estimation (MLE) to fit the whole distribution with an inverse gamma distribution (IGD) and obtain power-law decay exponents $\alpha + 1$ between $\sim 2$ and 3, consistent with the typical range found in the literature (Hovius et al., 1997; Malamud et al., 2004; Stark and Guzzetti, 2009; Frattini and Crosta, 2013). However, we note that at least three cases, B11, TW9 and C99, poorly follow an IGD, with a break in the distribution occurring at large areas, followed by a very steep decay.

When considering landslide estimated scar sizes, that is essentially a correction to reduce landslide polygon aspect ratio to 1.5, we observe a reduction of the largest landslide size by 2 to 10 fold, but a moderate reduction of the modal area. This is consistent with the fact that landslides with long runout distances are often over represented within the medium to large landslides (Fig Suppl 6). We also note that after the runout distances variability is normalized, the distribution of C99 agrees

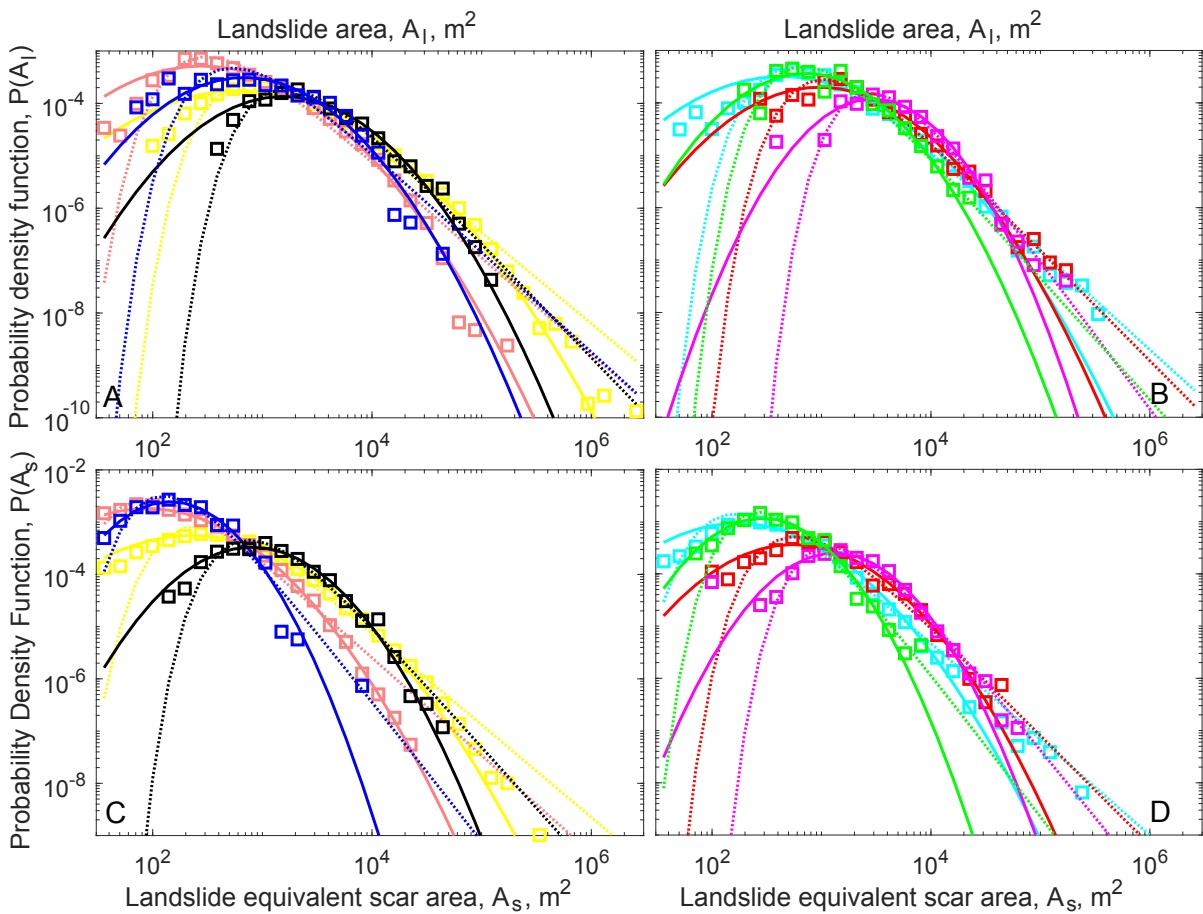

**Figure 3.** Probability density functions of landslide whole area (A, B) and estimated scar area (C, D). To improve visualization we split the 8 inventories in two groups. A Log-Normal and Inverse Gamma Distribution maximum-likelihood estimation for the whole distribution are shown by solid and dashed lines respectively.

better with an IGD. This is not the case however for B11 and TW9 that still feature a steepening of their distribution decay (and a divergence from IGD fit) beyond $\sim 10^3$ and $\sim 10^4$ m$^2$, respectively. Runout being normalized, this could be an artifact relating to residual amalgamation for TW9, but not for B11 where most landslides were mapped manually and amalgamation was avoided. In these two cases, for the whole landslide area or the landslide scar only, we note that a MLE fit of a log-normal
5 distribution agrees better to the data (based on the result of both the Kolmogorov-Smirnov and the Anderson-Darling test). For other inventories a log-normal fit is equivalent or worse than an IGD, but we note that the parameter describing the decay of both distribution are almost perfectly correlated (Fig Suppl. 7). Thus we take $\alpha + 1$ as a reasonable indicator of the relative proportion of large landslides within the different dataset and do not explore further the functional form of landslide size distribution and its implications, which we consider beyond the scope of this study.

### 3.1.2 Landslide and slope distribution

For all cases, we computed the frequency of slope angles above $5°$ based on the global 1 arc-second ($\sim 30$ m) SRTM digital surface model. In most cases, hillslopes have a distribution clearly independent from valley floors. However, for B08 and MI2 for example, the amount of plains in the study area do not allow to resolve the hillslope distribution. Therefore, for Micronesia we removed all slopes which are less than 10 m above sea-level, and for Brazil, we extracted the slope cells in the landslide distribution area but with a mask excluding the wide valley bottoms, allowing to obtain a hillslope distribution as an approximate Gaussian, with a mode significantly beyond our threshold of $5°$. To focus on the scar area of each landslide polygon, we extracted only the slopes for the highest elevation pixels representing a surface of $1.5W^2$. Then, we computed the probability density function for the landslide affected area and the whole topography (hereafter the "landslide" and "topographic" distributions) with a normal-kernel smoothing with an optimized bandwidth, as implemented in Matlab. We obtain topographic modal slopes, $S_M$, at $15.5°$ and $18.5°$ for the gentle landscape of Micronesia and Brazil, while in Japan and Taiwan we reach almost $30°$ (Fig 4A). The landslide distributions are unimodal, except for C15 that seems to have secondary modes at $S_M - 5$ and $S_M + 25$, and are systematically shifted towards steeper slopes.

To further quantify the differences in slope sampling between these events, we computed the ratio of probability between the slope distribution of the whole topography and of the landslide affected area only, $P_L/P_T$ (Fig 4). This ratio represents the tendency of landslide occurrence on a given slope to be more or less frequent than the expected occurrence of this given slope in the landscape. We refer to this as an oversampling or undersampling of the topographic slope distribution. To compare the events in different landscape we plot each event against $S - S_M$ (Fig 4B). An important issue is to determine whether the landslide probability can be considered a random drawing from slopes of the topography or not. Given that landsliding affect less than 10% of the landscape, the sampling of the topography by landslides can be approximated by a Bernoulli sampling. In this case, the central limit theorem gives the 95% prediction interval as $P_T \pm 1.96\sqrt{P_T(1-P_T)/N}$, with $N$ the number of independent draws, here taken as the number of landslide scars. The convergence of $N$ draws to $P_T$ within the prediction interval is only valid if $N > 30$, $NP_T > 5$ and $N(1-P_T) > 5$, implying that only very large samples can be interpreted towards the extremity of the topographic slope distribution, where $P_T$ is small.

For all events we observe that $P_L$ is significantly different from a random drawing of the topography with oversampling of the slopes beyond $S_M$ and undersampling below it (Fig 4B) . However, we note that for most events, the undersampling and oversampling is smaller than a factor of 2. Some cases (C15, J11 and TW8) have stronger oversampling ($> 4$) for $S - S_M > 25$ but they may not be representative ratios given the limited number of landslides and of slopes thus steep (i.e. $NP_T < 5$). The scars of C99 clearly departs from this behavior, with undersampling and oversampling of a factor of 10 and 6 at $S_M \pm 10$, respectively. B08 has also strong undersampling below $S_M$ but has a landslide distribution that rapidly converges to the topographic ones at high slopes.

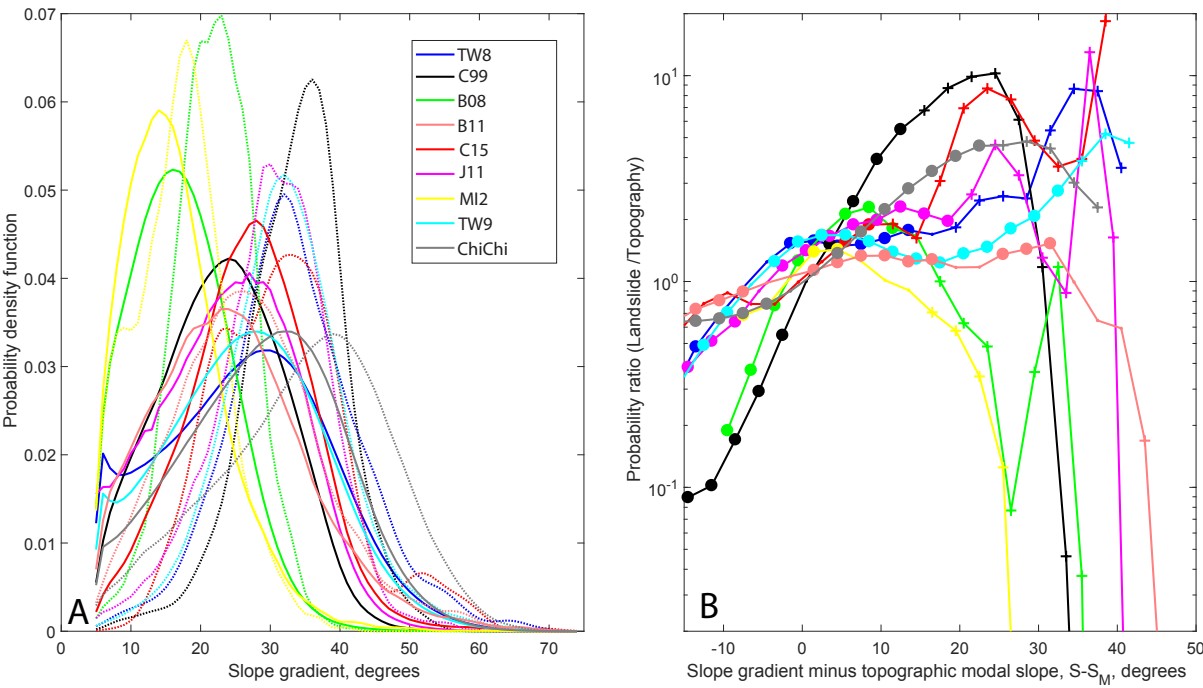

**Figure 4.** (A) Slope gradient probability distribution for the affected topography (solid) and the landslide scar areas only (dashed), for the eight rainfall events and as a comparison to the Chi-Chi earthquake. (B) Ratio of the two probability distributions against the difference between slope gradient and the modal topography. The ratios are estimated with the PDF averaged within 3° bins. Solid circles and dots represent ratios where the landslide probability is beyond or within, respectively, the 95% prediction interval of the topography distribution. Crosses indicate bins where data are insufficient for the validity of the Central Limit Theorem required to estimate prediction interval.

## 3.2 Correlation between rainfall metrics and landslide properties

### 3.2.1 Total landsliding

For the eight inventories, we observe a non linear increase of all metrics of total landsliding with the storm total rainfall (Fig 5). The increase is similar for the total area and volume, and best fit by exponential functions. We observe higher correlations with rainfall, when using the total scar area ($R^2 = 0.78$), estimated as $W^2$, instead of the total area ($R^2 = 0.72$). This is mainly due the very large reductions of area for C99 and B11 where long runout landslides were dominant (Fig 5). Correlation are generally higher with volume and also increase when we derive total volume from scar estimates (from $R^2 = 0.81$ to $R^2 = 0.87$). Note also that with these scar metrics, the relation to rainfall becomes equally or better fit by power-law function rather than an exponential function (Fig 5). This is because when including landslide runout the total landsliding of C99 and C15 is larger and creates an apparent asymptote, better fit by an exponential function. Last, we note that total volume values may change depending on which A-V scaling relations is used and with which assumptions, and their absolute values may be inaccurate

but this should not affect much the reported scaling form and exponents as potential biases should be relatively uniform.

Total number of landslides also tends to increase with total rain but the scatter is much larger (Fig 5). This is at least partly an artifact, given that for C99, MI2 and B11, high spatial resolution imagery allows to delineate many more small landslides and to mitigate amalgamation, whereas for B08, TW8 and TW9, the limited spatial resolution, the density of landsliding and our limited ability to split amalgamated landslides lead to an underestimation of the landslide number. Thus, even if landslide number may contain information, quantitative comparisons of the events are biased and we will not further interpret the total number of landslides in the following.

Last, we note that the landslide distribution areas (i.e., the regions within which landslides are distributed) also correlate strongly with the total rainfall. Only considering the eight inventories strongly suggest a power law form. However, based on the dataset reported for the Hurricane Mitch, the distribution area was at least 100,000 km$^2$, for maximum total rainfall at about 1500 mm (Cannon et al., 2001). Adding it to our fit, we found that power-law or exponential functions of the rainfall explained similar amount of the variance, 72% and 63% respectively.

In the next subsection, we compute landslide densities (in % of area), allowing to study the intra-storm variability of landsliding.

## 3.2.2    Maximum and mean landslide density

Understanding what controls landslide density is a key objective to better constrain hazards and their consequences. For each storm we compute the mean landslide density (in area and volume) by dividing total landsliding by the landslide distribution area (Fig 6A). This density represents the whole affected area and hides important spatial variability (Fig 1), thus we also compute the maximum landslide density, by computing the total landsliding (again in area and volume) within a moving window of 0.05° ($\sim 25$ km$^2$), assigning landslides to a cell based on their centroid locations, and selecting the maximal value (Fig 6B). Given the better correlation obtained above with a runout normalization, we focus on area and volume densities derived from scar estimates.

The mean landslide densities vary between $0.01 - 1\%$ and 100-10,000 m$^3$.km$^{-2}$ but with poor correlation with total rainfall ($R^2 = 0.01$ and $R^2 = 0.46$ for area and volume density respectively). Indeed, given that both total landsliding and distribution area increase strongly with total rainfall, their ratio is relatively independent. In contrast, the maximum landslide scar density and volume density range from 0.1 to 5% and 0.002 to 1.5 millions m$^3$.km$^{-2}$ , respectively, and are strongly correlated with a power-law of total rainfall ($R^2 = 0.76$ and $R^2 = 0.95$). We found very similar correlations when computing the local density on a grid of 0.03° or 0.1°, but degraded correlations when using the whole landslide area to compute landslide density ($R^2 = 0.40$ and $R^2 = 0.69$). We also note that, as for the total landsliding, maximum landslide density and volume density are significantly correlated with peak rainfall intensity, $I3$ ($R^2 = 0.58$ and $R^2 = 0.67$, respectively), and duration, $D$ ($R^2 = 0.70$ and $R^2 = 0.73$, respectively), although less strongly than with total rainfall.

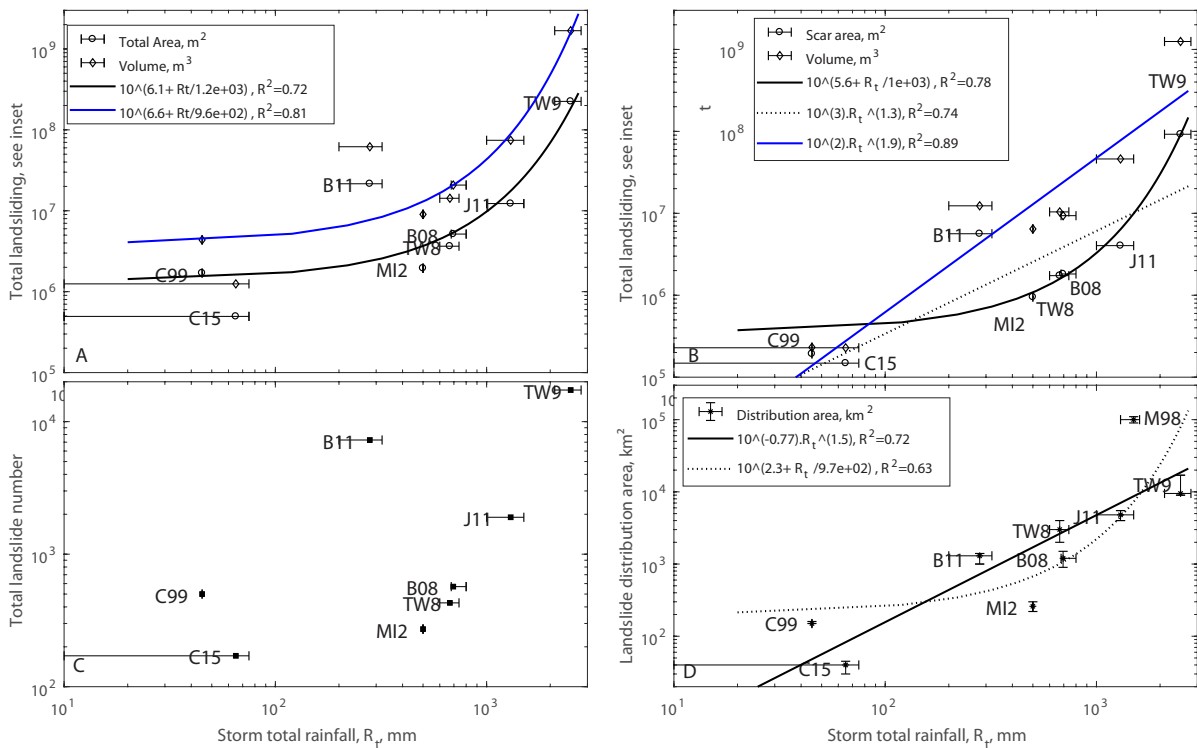

**Figure 5.** Total landsliding in area and volume derived from whole landslides (A) or from scar estimates only (B), total landslide number (C) and landslide distribution area (D) against storm total rainfall. M98 is for Mitch 1998. Horizontal error bars show a range of maximum storm rainfall when available (cf, Table 1). In A and B $1\sigma$ uncertainty on the total volumes and areas, ignoring potential landslide mis-detection (cf. methods), are smaller than the symbols ($< 10\%$). Vertical error bars are based on the range of affected areas in D, while we could not obtain quantitative uncertainties on the total number (C).

### 3.2.3 Landslide size, runout and position on slope

The decay exponents of the distribution of landslide area do not correlate significantly with any storm metrics (Intensity, duration or total rainfall) ($|R| < 0.1$). However, after runout normalization, the decay exponents of landslide scar area correlate with all metrics, although with significant scatter ($R^2 \sim 0.5$ Fig 3, 7). The two largest storms (J11 and TW9) have the lowest
5   exponents ($\alpha + 1 \sim 1.8$), and thus a large proportion of very large landslides, while the two smallest storms (C15 and C99) have a small proportion of large landslides and large exponents ($\alpha + 1 \sim 2.7$). However, intermediate cases are very scattered, as B11 and TW8 have similar total rainfall, peak intensity and duration but very different distribution with $\alpha + 1 = 1.9$ and with $\alpha + 1 = 2.6$, respectively. Still, randomly removing one event (i.e., jackknife sampling) we obtained $R^2$ between 0.4 and 0.7, with a similar mean $R^2$ about 0.5.
10  The decay exponents of the equivalent aspect ratio (Fig 3, Suppl. 6) do not correlate significantly with any storm metrics (Intensity, duration or total rainfall) ($|R| < 0.2$). Indeed, long runout landslides are abundant for the smallest storm, C99 and

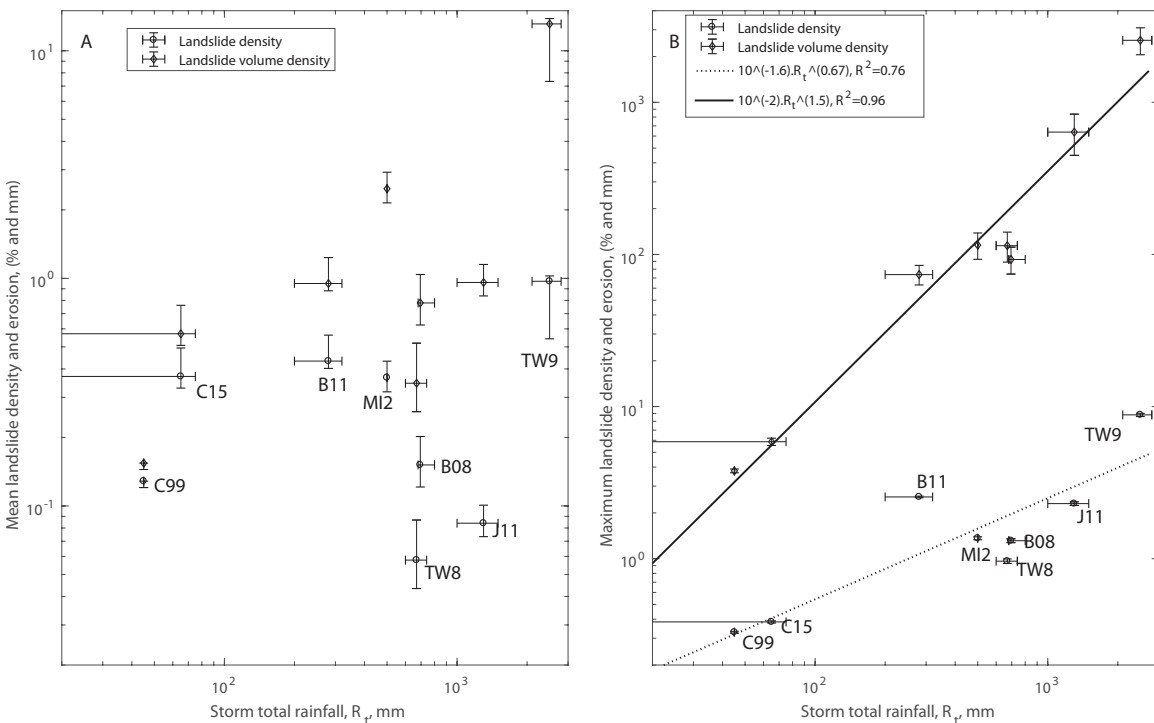

**Figure 6.** Mean landslide density (A) and peak landslide density in a 0.05° sliding window (B), against storm total rainfall. Landslide area and volume are derived from scar estimates, i.e., removing runout contribution. Horizontal error bars shows a range of maximum storm rainfall when available (cf, Table 1). Vertical error bars are based on the range of affected area in A (the most uncertain term), and represents $1\sigma$ uncertainty on the total volume and area density in B, ignoring potential landslide mis-detection (cf. methods).

C15, as well as for the second largest storm, J11, but are relatively rare for other storms (e.g., MI2,TW8, B08), spanning the whole range of storm indexes. Similarly, the mean or modal aspect ratio are similar for all event across all storm metrics, except for C99 which is heavily dominated by debris flow and has a modal aspect ratio $> 10$.

We have observed that almost the eight events behave similarly with respect to the distribution of topographic slopes, not

5 suggesting strong link with the individual storms properties. The C99 event has a different behavior that may relate to the fact that it was the shortest storm with the smallest total, or that it was the only cases occurring in high elevation terrain, with sparse vegetation. C15, the second shortest and smallest storm event may also have strong oversampling about 20° beyond $S_M$ but the limited number of landslides does not allow to confirm the significance of this oversampling.

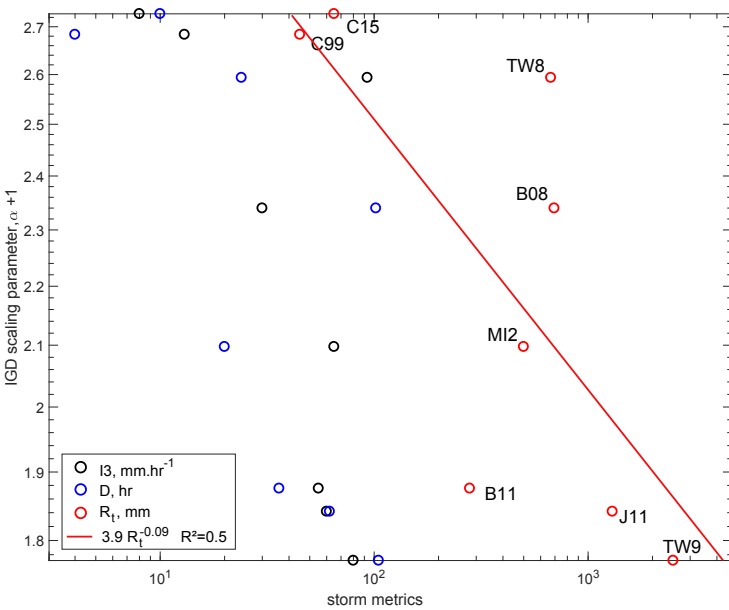

**Figure 7.** Landslide scar size distribution decay exponents against storm total rainfall (red), storm duration (blue) and storm peak intensity (black). The best least-squares fit is shown in red. The reduction of the decay exponents with increasing storm magnitude indicates an increase in the proportion of large landslides relative to small landslide.

## 4 Discussion

### 4.1 Scaling between rainfall and landsliding

We found that total landsliding, peak landslide density and the distribution area of landsliding were all best described as increasing as a power-law or exponential function of the total storm rainfall, $R_t$. Our mechanistic understanding of landsliding predicts that, for a given site, the driving force leading to failure is the reduction of normal load and friction due to the increasing pore-water pressure (Iverson, 2000). This requires progressive saturation of the material above the failure plane, and depends directly on the total amount of water poured on the slopes. However, we can envision that landscapes may rapidly reach an equilibrium in which all unstable slopes under rainfall conditions frequently occurring would have been removed. In this framework, the rainfall amount relative to the local climate would be more relevant than absolute rainfall, requiring an analysis in terms of deviation from the mean rainfall or in terms of rainfall percentiles (e.g., Guzzetti 2007). Although we could not define rainfall percentiles in each area, we note that normalizing $R_t$ by the mean monthly rainfall relevant for each storm, we still find a decent correlation with the peak landslide density, implying climate normalized rainfall variable may be driving landsliding (Fig Suppl. 8). The antecedent rainfall is also expected to play a key role, in controlling the saturation level before the triggering storm (e.g., Gabet et al., 2004; Godt et al., 2006). However, if the regolith is already close to the field capacity, significant parts of antecedent rainfall may be drained from the regolith within some hours or days (Wilson and

Wieczorek, 1995), and as result, the contribution of past storms may be negligible compared to heavy rainfalls over relatively short time intervals (1-4 days). However, for moderate storms, like C15 or C99, and especially during dry periods when the slope is saturated below field capacity, the role of antecedent rainfall may be more substantial. Thus, we expect that moderate storms happening after prolonged dry or wet periods may deviate downward or upward from the scaling, respectively. We also note that an abundance of larger and deeper landslide strongly influencing the total volume or erosion, may depend on deeper water level rather than regolith saturation and thus may be most sensitive to water accumulation over several days rather than a few hours (Van Asch et al., 1999; Uchida et al., 2013). Therefore, although we obtained a good correlation without considering antecedent rainfall, its role should be assessed in future refined scalings. Last, the scaling reported here is based on events where all landslides occurred within a short time frame (few hours to few days), and would not apply to a monsoon setting where landslides occur more or less continuously during several weeks (Gabet et al., 2004; Dahal and Hasegawa, 2008), driven by continuous, heavy but unexceptional, rainfall. Indeed, in a long period with fluctuating rainfall such as the monsoon, drainage and storage of water will certainly not be negligible and the derivation of a soil water content proxy will be necessary (e.g., Gabet et al., 2004).

The strong correlation between $R_t$ and $A_d$, suggests that storms able to generate greater amounts of rainfall also tends to deliver a sufficient amount of rain over broader areas. For tropical storms and hurricanes (5 out 9 cases in Fig 5D) a number of studies (cf., Jiang et al., 2008, and references therein), found that the maximum inland storm total rainfall (i.e, $R_t$ for us) correlated well ($R > 0.7$) with a rainfall potential defined as the product of storm diameter and storm mean rainfall rate within this diameter over storm velocity, each terms measured 1-3 days before the storm made landfall. It was also generally observed that rainfall intensity is higher closer of the storm core, thus potentially tightening the link between $R_t$ and a given storm radius with intense rainfall and high landslide probability. These observations would imply linear proportionality between $R_t$ and $A_d$ and could be consistent with the observed power-law trend (exponent 1.5) ( Fig 5), especially if some further links between $R_t$ and mean storm intensity or velocity exist. Potential links between $R_t$ and $A_d$ for smaller scale storms (C99,C15, B08 and B11) are harder to interpret, and we cannot exclude that it is a coincidence allowed by our small number of events. In any case the broader zone is not likely to receive homogeneous rainfall amount, decoupling mean landslide density from storm maximum strength (Fig 6A). The variability of rainfall within these extended zones is likely a main control on the spatial variability of landslide density, although lithological properties or slope distribution may also matter. Indeed lithological boundaries, or a lack of steep slopes can sometimes explain spatial variability in landsliding, but not all of it (e.g., Fig Suppl. 9). In any case, it seems clear that to predict the spatial variability of landsliding, the rainfall spatio-temporal pattern is a primary requirement. The good correlation between storm total rainfall and peak landslide density is encouraging and suggests that, as most mountainous regions may have sparse instrumental coverage, the use of satellite measurements (Ushio et al., 2009; Huffman et al., 2007) or meso-scale meteorological models (e.g., Lafore et al., 1997) may be required to understand the spatial pattern of rainfall induced landsliding.

A few non-linear scaling between total landsliding and total rainfall have been reported at the catchment scale, but were derived from datasets not easily comparable to the one presented in this study (Reid, 1998; Chen et al., 2013; Marc et al., 2015). The details of this scaling is of importance in order to understand the impact of extreme rainfall events and more generally which

type of rainfall event contributes most to sediment transfer over long time scales (Reid and Page, 2003; Chen et al., 2015). We also found non-linear scaling between $R_t$ and total landslide area, but without a strong statistical difference between exponential or power-law. Exponential functions yield a minimum landsliding amount at low rainfall, that is not physical justified. This apparent contradiction may, however, be resolved by considering a rainfall threshold below which landsliding is null. The

higher correlation between $R_t$ and total volume, is likely due to the fact that $R_t$ correlates very well with maximum landslide size ($R^2 = 0.8$ with whole landslides, $R^2 = 0.9$ and almost linear correlation with scar estimates, Figure Suppl 10), with large landslides contributing most of the total volume and erosion. A correlation between $R_t$ and large landsliding may arise because landslide stability is determined by the ratio between pore pressure and the total normal stress on the slip plane, meaning that larger landslides that have usually deeper failure planes (Larsen et al., 2010), may only fail with greater precipitation amount.

However, given that the trend between total rainfall and the landslide size distribution is much weaker, this correlation may also partly result from a sampling bias as the probability to draw large landslides increases with the total number of landslides. For now, our unreliable estimates of total landslide number do not allow to quantify this effect.

In any case, several caveat should be taken with the preliminary scaling between total storm rainfall and total landsliding. First, the definition and limit of a single "storm" is not generally agreed in the meteorological community, because the atmospheric

fluids suffer perturbations with scale interactions, and therefore with events not independent from each other. Ideally, future studies could categorize storms according to some space-time filtering and analyze the scaling with total landsliding for each storm category. Currently, our database is not sufficient for this. Second, linking total rainfall in a limited area and the total landsliding within the storm footprint implicitly suggests that storm rainfall is somewhat structured with some internal correlations between peak rainfall, storm size, and the spatial pattern of rainfall intensity within the storm. This seems to be the

case for large tropical storms (Jiang et al., 2008), but should be explored for a broader range of storm types. Orographic effects (e.g., Houze, 2012; Taniguchi et al., 2013), focussing high intensity rainfall on topographic barriers, may also enhance such correlation between local total rainfall, and the broader pattern of rainfall and landsliding. Last, the scaling with rainfall may also be obscured by outliers due to processes not controlled by rainfall. For example, the inclusion of the very long runout components in several inventories led to larger scattering for both power-law and exponential models and to favor the latter.

Therefore, the proposed runout correction seem essential for future studies. Another issue concerns the normalization of landscape parameters affecting the susceptibility to landsliding, such as hillslope steepness and mechanical strength (Schmidt and Montgomery, 1995; Parise and Jibson, 2000; Marc et al., 2016). Nevertheless, the proportion of flat or submerged land within the area of the most intense rainfall must limit the total landsliding, as it was certainly the case for MI2 or B08 (Fig 1). Recent, widespread antecedent landsliding may also reduce subsequent susceptibility to rainfall triggering by removing the weak layer

of soil or regolith on steep slopes. In the pre-event imagery we did not see specific evidence of such limitation, except maybe for J11, where abundant pre-event fresh landsliding were visible near 136,25°E/34,29°N and 135,9°E/34,20° and very few new landslides occurred. More systematic evaluation of this effect may be important when comparing quantitatively landslide and rainfall pattern. In any case, it is clear that further analysis of this database, possibly extended with additional landslide inventories, should be used by future studies to refine the scaling with rainfall and incorporate the effects of controlling parameters

such as available topography, antecedent rainfall or regolith properties (e.g., strength and permeability).

## 4.2 Relation between rainfall and landslide properties

We found an increase in the proportion of large landslide scars with all storm metrics, but clearest with the total rainstorm (Fig 7). This is consistent with the idea that large landslides require larger amount of rainfall to be triggered (Van Asch et al., 1999), as discussed above and exemplified with the strong correlation between $R_t$ and maximal landslide scar (Fig Suppl. 10). The large remaining scatter suggests that other differences between the inventories matter, such as differences in the mechanical properties of the substrate (e.g., Stark and Guzzetti, 2009). Indeed, large lithological contrast exist between each events, and sometimes within event (Fig Suppl. 9). The variability in extent and thickness of weak superficial layers (i.e., soils) between the different landscapes affected may also be important. Variations in slope distribution and relief are also wide between each case (Fig 1, 4) and have also been reported to influence landslide size (Frattini and Crosta, 2013). We note that the correlation between peak rainfall intensity is opposed to what could be expected, as more small landslide are expected for pulses of very intense rain leading to the occurrence of transient high pore-water pressure pulses at shallow depth (Iverson, 2000). Given that water retention and hydraulic conductivity may easily change by orders of magnitude between different environments, it may be needed to normalize intensity by the regolith hydraulic conductivity (Iverson, 2000) to understand its potential influence. For the moment we consider that the correlation between $D$ and $I3$ and the landslide size distribution exponents likely arises because of the correlation between thes storm metrics and $R_t$ (Fig Suppl. 4) In any case, our results suggest that it is not only the landscape properties that set the landslide size distribution, but also the trigger characteristics, as previously reported for earthquake induced landslide size distributions (Marc et al., 2016). This means, for example, that the influence of forcing variability should be assessed and normalized before inverting landslide size distribution parameters to obtain regional variations of mechanical properties (e.g., Gallen et al., 2015). In contrast, aspect ratio or runout did not correlate well with storm metrics and thus obscured any direct correlation between storm metrics and the decay exponents of whole landslide area. This underlines again the importance to isolate scar geometry to deconvolve processes driving landslide initiation and landslide runout. As for the landslide size distribution, landslide runout may likely be influenced by slope and relief distribution, as well as by hydrologic processes. The case of C99, with exceptional runout for most of its landslides is interpreted has the effect of a low infiltration rates favoring large runoff generation (Godt and Coe, 2007). This may also explain the abundance of debris flow in other places (C15, J11) but cannot be verified without information on infiltration rate in these places to normalize the intensity variations. An alternative could be to study various storms occurring over the same region, and where infiltration rate or conductivity could be assumed constant, for example with datasets from multiple typhoons in Taiwan (Chen et al., 2013). Finally, we observed that most rainfall induced landslide inventories sample the topographic slope distribution with a minor oversampling beyond the topographic modal slope (Fig 4). This is in contrast with the case of earthquake triggered landslides, where we systematically observe preferential landsliding on the steepest slopes (Parise and Jibson, 2000; Gorum et al., 2013, 2014)(Fig 4), quite similar to the case of C99. One dimensional static force balance shows that the steepest slopes are the most unstable, and therefore the over-sampling of steep slopes must be expected if the forcing (pore-water pressure or shaking) is randomly distributed across the whole topography. To obtain equal sampling or under-sampling of steep slopes, the forcing intensity must be anti-correlated with slope gradient. Rainfall may be mostly independent of local slopes, but probably not the

pore pressure rise that depends on the underground water circulation and thus topography. The pore pressure will thus depend crucially on vertical infiltration and drainage, but also on along-slope contributions. For example under moderate intensity, but long rainfall, pore-water pressure will reach higher level in concave, downslope areas (Montgomery and Dietrich, 1994) with relatively large drainage area, and thus lower slope gradient (Montgomery, 2001). In such view, landslide slope statistics would bear information on the type of rainfall, short and intense (relative to local permeability) for steep over-sampling, while equal sampling and under-sampling would relate to moderate and long rainfall. This framework might explain the preferential location on steep slopes observed for the very short duration C99 and possibly C15 (Fig 4). However, the statistics of C15 are weak and C99 strong oversampling may relate mainly to specific mass movement triggered by surface runoff such as rilling and firehose (cf., Godt and Coe, 2007). These processes also require high intensity, short duration events, but also low surficial infiltration rate leading to overland flow able to mobilize relatively loose surface materials. For other events, we analyzed the slope-gradient-drainage area relationship for topography and landslide subset and did not find clear over-sampling of high-drainage and gentle gradient areas in the landslide distribution. It is well possible that a 30 m DEM is not able to resolve accurately the fine-scale pattern of slope and drainage on the hillsides, where landslides occur, but it may also suggest that the upslope drainage area is not the main explanation. For example, the subsurface drainage efficiency may also increase with slope gradient, thus making very steep areas less likely to develop large pore pressure and possibly explaining the preferential landsliding of slopes just above the modal slopes for almost all events, indepedent of rainfall properties. Hydro-mechanical modeling at the catchment scale (e.g., von Ruette et al., 2013), applied on several of our dataset may be the only way to test between these different hypothesis. Further constraints on the processes controlling rainfall-induced landslides may also be achieved through a discussion in terms of relative distance from ridge and river (cf, Meunier et al., 2008), as intense and brief storms should yield uniformly distributed landsliding, in contrast to longer, less intense storms favoring near river slides.

## 5   Conclusions

We present landslide inventories (comprising from a few hundreds to more than 15,000 polygons) associated with eight triggering rain storms from Asia, South America, and North America. We hypothesize that these datasets constitute a global database of rainfall-induced landslides, that allows studying a number of landslide metrics and their relations to rainfall and landscape properties. Indeed, although spanning a large range of landscape settings, whether in terms of topography, climate or vegetation, the magnitude of landsliding scales non-linearly with the magnitude of the storm, here quantified with estimates of the total rainfall. We also found that correlation between landsliding and rainfall is higher when considering landslide scar estimates obtained through a normalization of landslide runout, as the runout distribution does not clearly correlate to rainfall. Therefore, after removing the runout contribution (i.e., focussing on scars) we also find that landslide size distribution decay exponent seems to be partly controlled by totall rainfall, with a greater proportion of large landslides for larger total precipitation. This implies that variations of landslide size distribution cannot be directly interpreted as variations in landscape properties. For total landsliding and maximum local landslide density, power-law scaling based on total rainfall explains 74% (87% for total volume) and 76% (95% for volume density) of the variance, respectively. Adding a number of other storm events as

well as integrating other rainfall forcing parameters or landscape susceptibility properties have therefore the potential to yield robust prediction on the magnitude of rainfall induced landsliding. Last, we identify that compared to earthquakes, storms tend to trigger landslides that only slightly oversample the topographic slope distribution, possibly due to faster drainage on steep slopes or to underground water accumulation on high drainage-low gradient portions of the hillslope. This may bring new, although less straightforward, implications for the difference in resulting topography of bedrock landscape dominated by rainfall-induced or earthquake-induced landslides (Densmore and Hovius, 2000). Although preliminary these insights and scaling clearly show the value of mapping systematically a large sample of the landslides that can be related to a single storm and we identified a number of recent storm events were such type of inventory could be produced. Although not systematically addressed here, landslide density spatial pattern is likely strongly related to the spatio-temporal pattern of rainfall, and constraining the quantitative links between the two is another challenge that may be addressed with some of the inventories presented here. More generally, the database presented here may also serve as a benchmark for developing and comparing rainfall induced landslide models, whether empirical, semi- or fully-deterministic. These future developments are important challenges in order to understand the natural hazards posed by rainfall-induced landslides as well as their specific implication for the erosion and topographic evolution of landscapes in different climatic settings.

*Acknowledgements.* The authors are grateful to the thorough and constructive reviews from Dave Milledge and an anonymous reviewers that helped to improved and clarify considerably this manuscript. OM thanks Patrick Meunier for discussions on prediction interval and the statistical significance of probability distribution ratios. This work was carried with the support of the French Space Agency (CNES) through the project STREAM-LINE GLIDERS "SaTellite-based Rainfall Measurement and LandslIde detectioN for Global LandslIDE-Rainfall Scaling". Additional support by the Open Partial Agreement "Major Hazards" of Council of Europe through the project "Space data in Disaster Risk Reduction: using satellite precipitation data and hydrological information for leapfrogging landslide forecasting" was available.

*Data availability.* All imagery used to map the landslides is available in public repository except the images from Formosat-2, Geoeye-1 and airphotos used for the events in Colorado, Micronesia and Japan. For the two former, landslide maps are directly available from the USGS at http://pubs.usgs.gov/of/2003/ofr-03-050/ and https://pubs.usgs.gov/of/2004/1348/. For Japan and Brazil, extensive high resolution imagery is available over most of the areas of interest in Google Earth. The new digitized landslide inventories are available upon request.
The authors acknowledge support from the Typhoon and Flood Research Institute, National and Applied Research Laboratories which provide rain gage information in Taiwan from the Data Bank for Atmospheric and Hydrologic Research service (https://dbahr.narlabs.org.tw). In Japan, rain gage data was accessed thanks to Japan Meteorological Agency. The authors gratefully used the global GSMaP rainfall products (http://sharaku.eorc.jaxa.jp/GSMaP_crest/) and SRTM-30 m DEM provided by USGS https://gdex.cr.usgs.gov/gdex/

*Author contributions.* OM designed the study and performed all analyses with some inputs from AS, JPM and MG. TU provided landslide and rainfall data for the J11 event. SHC provided imagery and landslide data for the TW9 event. OM wrote the manuscript with contributions from all co-authors.

*Competing interests.* The authors declare no conflicts of interest with the present study.

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
