# Peer review of "Initial insights from a global database of rainfall-induced landslide inventories: the weak influence of slope and strong influence of total storm rainfall"

_Earth Surface Dynamics, 2018_

## Referee Comment (RC1) · D. Milledge (Referee) · 5 May 2018

Review of Marc et al. "Towards a global database of rainfall-induced landslide inventories:

first insights from past and new events" by David Milledge.

**Major Comments**

This is a well executed study with novel and interesting findings. I have three general comments and a large number of minor comments but neither the major nor minor comments reflect a fundamental problem in the research in my view.

I am not convinced that it is essential (or helpful) to present your inventories as the only inventories that are suitable for this type of analysis (as you seem to do on P2-3). Instead you could simply say they are one set of inventories and they demonstrate the power of this type of approach. I am not convinced of the need for landslides beneath an entire storm footprint to be mapped and am sceptical that entire storm footprints can be convincingly defined so I'm not convinced by your critique of studies that analyse far smaller study areas (other than on sample size grounds).

The methodology description could be more consistent between inventories. Similar information is reported for each case but the style of the reporting differs and some key information reported in some cases is not present in others (e.g. image source, image resolution, acquisition date).

I am not convinced that your focus on 'comprehensive' inventories is necessary nor that examination of total landslide numbers, volumes or areas are particularly meaningful in relation to rainfall triggered landslide inventories (though I think the findings on landslide density and slope are extremely interesting and thought provoking). This focus might reflect a desire for comparability to co-seismic landslides but I think the two triggers are importantly different. For example, it is extremely difficult to define the spatial and temporal limits on a single storm. In addition I find the results relating to total numbers, volumes and areas less convincing because they are predicted from a small number of point rainfall records. A clearer explanation of why 'comprehensive' inventories and total statistics are important would be a valuable addition to the paper.

**Minor Comments**

P1L4: Associated to: How do you know that these events are associated to one another.

P2

L8: deterministic approaches inapplicable: I think this statement is a little strong. Is it really fair to say that they are inapplicable given their data requirements.

L30: comprehensive: this term needs defining.

L34: Why is it insufficient? I think you need to demonstrate this. Is this a sample size argument? Some things won't be possible to calculate but others will. What can you and can't you do with a subset inventory and how big does the subset need to be?. Is it ever possible to capture the full inventory for a storm? How do you define its bounds?

P3

L4: landslide scale: I am not clear what this means. Could you define it?

L7: comprehensive mapping: where do you start and finish. Your definition of a storm is very important here and I don't see it at the moment. For example shouldn't the Morakot mapping extend to the Phillipenes and China on this basis?

L14: adequately: how do you quantify adequate representation, what would inadequate representation look like and how do you know whether a representation is adequate?

L30: this gets at a difficult issue, what do you include as a landslide? I think you need a clear definition that can be applied across all inventories and I don't see one at present. Divergence from the definition in different inventories will introduce bias to your results.

P4L4: whether or not the statistical properties of a subset are representative: you need to demonstrate that they are not representative for your argument here to hold and it is not obvious that this is the case. They might not be representative because of the sample size but why should you need the all landslides triggered by a particular storm, it seems reasonable to assume that one catchment is independent of another for these processes and on these timescales.

P5

L15-19: why take this approach rather than breaking up the multi-headed polygons manually?

L6-17: this methods description is difficult to follow. Image acquisition dates and image resolution information is missing in some cases. It would also be useful to give some indication of the performance of the automated classification with respect to manually mapped landslides.

P7

L1: there is quite a long window between pre and post event imagery in some cases. How confident can you be that another storm did not trigger some of the landslides? What evidence do you have that this is the case?

L2: mapped automatically: I think you need to include a methods description for this automatic mapping and some information on how the quality of this mapping was evaluated.

L4: fluvial system: How do you define the fluvial system and how did you identify it for the study area?

L5: to map at least the largest: I don't understand what this means, were there some areas of your study area that you did not have high resolution imagery for? If so what fraction of the study area was this and what impact does this have on the inventory as a whole?

L12: Specific dates are missing for the Landsat images. This is a 2 year window, which seems a very long time. How confident can you be in assigning landslides to a single event within that window and what is the basis for this confidence? This is particularly important given your earlier critique of other inventories.

L20-21: maximal forcing: this doesn't seem to be consistent with your argument for the importance of complete landslide footprints. You are comparing the forcing at a single location within the footprint to the properties of the entire footprint.

L30: landslide densities: calculated over what window size, I think that this choice will be critically important. On a small window density will have multiple local peaks.

L33: why not use 3 gauges for Colorado? Where were the next nearest gauges and why were they discounted?

P8L11: continuous period: I'm not totally clear what this means, does it mean that if there was no rain in a 3 hour period then that is the end / start of the storm? Was the same duration criteria applied to all records?

P9

L5: how, and where, did you measure landslide width?

L6: I think you could state this more simply by saying that you assume that scars have equal length and width. This is the same assumption used by Pelletier et al., 1997.

P10

L23: isolated remote landslides: how were these defined?

L24: to what extent is the landslide distribution area constrained by your study area (i.e. the extent of available images). Taking this to an extreme did Typhoon Morakot trigger landslides in China or the Phillipenes and should these also be included? This again reflects something that I think you need to discuss somewhere, the differences between rain storms and earthquakes as triggers: where are they similar enough to borrow frameworks from one another and where do they differ?

P11

L3: typically have power law: they have typically been fit with these distributions but do we know that they typically follow that distribution or do we fit power laws tailed distributions without testing alternatives (e.g. log-normal).

L19: must also: must is a strong statement, could it alternatively be due to different mapping criteria?

L24: peculiar distributions: are these distributions peculiar if you are seeking power laws but not if other alternatives are considered? Have you tried a log-normal distribution? Negative curvature of the tail in log-log space sometimes indicates better fits for log-normal distributions?

L26: Why use a least square fit to represent the power law tail? The problems associated with using least squares fits to binned data rather than an MLE have been widely discussed (e.g. White et al., 2008; Clauset et al., 2009) and Clauset et al. (2009) provide appropriate tools to fit only the power law tail using an MLE.

L29: aspect ratio below 2: why below 2? What are the specifics of the equation? I had understood it to be A=w^2, which would give an aspect ratio of 1.

P12

L4-8: Why is this censoring of low slopes necessary? I am not clear on what you are trying to achieve by removing them?

L8-10: generating a histogram then smoothing it seems an unusual approach to this problem, results will likely be sensitive to both the smoothing window and smoothing function. Given the theoretical basis for Kernel density estimation (e.g. Cox, 2007), why not use this approach?

P13

L5: initiation point: I don't think you have previously defined this or explained how these points are identified.

L11: focussing on scar areas seems sensible but this particular approach seems strange and the choice of modal topographic slope somewhat arbitrary, could you provide a more robust explanation for this choice? Alternatively couldn't you have used your previously defined scar area (w^2) to identify scars as the highest w^2 area of each polygon? This would be consistent with your previous definition and would avoid introducing an arbitrary slope threshold which could bias the results.

L14: Could you use line thickness to indicate the slope beyond which small numbers of cells in the value range preclude interpretation of the line? It would be useful for the reader to know where that point is for each dataset. Also could you colour the lines in Fig 5 by storm duration? This might make it easier to pick out the behaviour you are identifying in the text and to make a connection between 5A and 5B.

P17L9: Total storm rainfall: These results are extremely interesting. They suggest that absolute rainfall properties are good predictors for landslide properties. In the rainfall threshold literature there has been debate over whether absolute rainfall properties are driving failure or whether it is the degree of deviation from normal conditions (e.g. expressed as percentiles). It might be useful if you could reflect on this in relation to your findings. Would a plot of rainfall percentiles for these storms look very similar to the plot of absolutes that we see here?

P18L17: we have no clear physical explanation: isn't this something that either extreme rainfall community or the hurricane community have thought about? It would be useful to point readers to key reference from that literature here even if you don't strongly back one particular explanation.

References

Clauset, A., Shalizi, C.R. and Newman, M.E., 2009. Power-law distributions in empirical data. SIAM review, 51(4), pp.661-703.

Cox, N.J., 2007. Kernel estimation as a basic tool for geomorphological data analysis. Earth surface processes and landforms, 32(12), pp.1902-1912.

White, E.P., Enquist, B.J. and Green, J.L., 2008. On estimating the exponent of power-law frequency distributions. Ecology, 89(4), pp.905-912.

**Typographic errors and wording suggestions**

P1

L10: storm -> storms

L15: rainfalls -> rainfall

P2

L7: At this day -> at present

L22: allow to make -> allow

L23: region -> regions

L23: (large slope -> (hillslope

L26: These progresses have - > this progress has

P3

L7: delete: , and thus

L15: inventory > inventories

L17: it isn't clear what you mean here, perhaps add: size (total area), geometry (length, width and depth) etc.

L25 were > was

P4

L15: N-s should be Ns

L34: avoid > avoids

P5

L2: twice more: or twice the number (i.e. 3n or 2n)?

L31: you use a variety of date formats which is a little confusing.

L34: dates are needed for the FORMOSAT-2 image acquisition.

P6: letters missing from Fig 1. Colours of landslides are very difficult to distinguish.

P7

L23-27: I don't think this is relevant here, I suggest moving to the discussion.

L30: average record properties: it isn't immediately clear what you mean here.

L33: closest of > closest to

P8

Table 1 caption: Reference are as follow > References are as follows

L10: other > over

L20: is > are

P9

L1: polygons > polygon

L3: allows > allows us

P10

L25: the built > the

L26: 0.2 and > 0.2 to

P11

L10: with important total precipitation: this doesn't seem the right set of words

P13

L13: artifact due to > artefacts of

P16

L4: S??: figure details missing.

P18

L23: prime > primary

P21L5: storm tends > storms tend

---

## Author Comment (AC1) · 6 May 2018

I thank warmly David Milledge for its in depth lecture and his many thoughtful and constructive comments.

We will re-evaluate the novelty and interest of "comprehensive" inventories and further discuss and nuance some sections of the manuscript to answer these points.

We will also clarify different point of the methods, in terms of inventory mapping, geometry correction or statistic robustness of the landslide slope probability.

We will answer these points and the detail comments exhaustively when receiving the

comments from the other referee.

**ESurfD**

---

## Referee Comment (RC2) · Anonymous Referee #2 · 16 May 2018

General Comments

In this manuscript, the authors Marc et al. seek to understand what governs the spatial and geometric characteristics of rainfall-induced landslides that result from single storm events. Toward that end, they compile a database of landslide inventories from single storm events spanning the past twenty years. The database is well considered and thorough, and the strengths and limitations of each are discussed (e.g., availability of local rain gauge data, etc.). With this dataset, the authors then compare landslide spatial characteristics (number of landslides, area affected by landslides, landslide density) and geometric characteristics (landslide total area, landslide scar area)

[Figure]

with precipitation characteristics (rainfall duration, storm intensity, total rainfall). In this analysis, the authors make a number of interesting findings. For example, they find that the longer-duration storms result in an increasing number of lower-gradient land-slides. Additionally, they show that while landslide volume and spatial density vary as a nonlinear function of storm total rainfall, other landslide parameters do not appear to depend on storm characteristics such as rainfall intensity. From this analysis the authors conclude that their global inventory of single-event rainfall-induced landslides can be queried to answer fundamental questions about the spatiotemporal evolution of landslides in response to hydrologic forcing.

Understanding at a broad scale how a landscape may respond to a given storm event is fundamentally useful for both geomorphic and hazard assessment applications, and this topic should appeal to the readership of ESurf. The rigorous dataset compilation by the authors provides a sound basis to start better quantifying these relationships, and the statistical methods applied to the dataset are well founded and consistent with those used by the community and should be readily reproducible by other researchers (assuming the database will be available online). The observed correlations between storm magnitude and landslide area draw an interesting parallel to empirical studies of coseismic landslides (e.g., Keefer, 1994), and the data support the authors' conclusions both that rainfall magnitude (expressed as total storm rainfall) is a good indicator of landslide hazard potential and that the increase in lower-sloped landslides occurring over longer duration storms may reflect timescales for water to infiltrate lower portions of the landscape. Although the idea of rainfall-induced landslides occurring in lower-sloping sections of a hillslope is well established (e.g., Reid and Iverson, 1992; Densmore and Hovius, 2000), this work shows that the prominence of this effect potentially depends on the storm magnitude and duration.

In terms of the general aspects of the manuscript presentation and layout, I find that the paper needs some fine-tuning and clarification, but overall it is close to being a finished product. The abstract provides a clear summary of the work, and the overall

structure and segmentation of the manuscript is easy to follow. The title largely makes sense, although I find the last section stating "first insights from past and new events" confusing since all datasets are within the past twenty years and the youngest event occurred in 2015, and I'm not sure what that phrase adds to the description of the research. There are a number of places where language needs to be altered slightly, and I've tried to provide examples below in the technical comments section of the review. Mathematical formulae appear to be largely correct, but abbreviations for the landslide inventories (although intuitive) are not defined before they are used. Additionally, I found that Figure 2 should be modified, as it is very difficult to see the pink landslide polygons draped over the red and green topography. I imagine this would be especially difficult for people who are red-green colorblind. The supplementary material complements the manuscript well, and I have a few comments regarding supplementary figures below. Although I have a few additional concerns related to content and clarification, overall I think this paper will make an interesting contribution to ESurf.

Specific Comments

1st Paragraph: I'm not sure I agree with the statement that the goal of constraining quantitative relationships between landslide occurrence and rainfall is out of reach. The authors cite examples of this in the same paragraph. I do agree that there is certainly room for improvement in his area, which I think is the implied sentiment here.

Equation 1. I appreciate that the authors' goal here is to try to bridge the gap between purely deterministic models and purely statistical models, but I think that there needs to be a little more clarity. At the end of Paragraph 1, for example, you state that certain parameters such as permeability and cohesion that are required for deterministic approaches make a landscape-scale approach in data-poor regions inapplicable, yet you specifically include those parameters in your idealized semi-deterministic Equation 1. Why then is a deterministic approach not appropriate? I think that a bit more discussion

might clarify these discrepancies.

Lines 31-34: I'm confused by this sentence. When specifically are data from 2010-2012 used? When May-June 2009 data are not available for a specific location in the landslide-affected area?

Section 2.2 overall.

I'm also confused with the general methodology here. You map landslides on 30 m Landsat imagery, as well as on higher resolution imagery within Google Earth, but only in areas where a negative change in NDVI was observed at the 30 m scale. You then say that field mapping in the area reports twice as many landslides than was observed via remote sensing, but that the missing landslides must be smaller than ~1 m resolution. Could the missing landslides not just be in areas that didn't result in a negative NDVI shift in the landsat imagery? For example, a small translation or slump in a forested area may not affect a 30 m pixel.

Lines 23-27: In your discussion of peculiar landslide frequency distributions, you focus on deviations from the (perhaps) expected Inverse-Gamma distributions at the large end of the distributions. What about deviations on the smaller end? For example, in the Total Area distributions, TW9, B11, C99, and J11 deviation from the maximum likelihood estimations pretty substantially for small landslide areas. Converting total landslides area to landslide scar area (As), the TW9 distribution especially deviates quite far from the expected P values. Is there a known reason for these deviations? I am far from an expert on landslide frequency distributions, but it seems worth discussing since it is quite apparent on Figure 3!

Figure 3: Similarly, I don't believe it is mentioned why the authors choose to break up their landslide populations into two groups. Is this just to more easily visualize? Or is it

based on the quality of datasets?

Lines 11-12: Is there a plot that shows the relationship described on these lines? I couldn't find one. Maybe it would be worth including these in the supplemental material.

Lines 8-9: Very cool.

Lines 12-15: If there is a continuous forcing of heavy rainfall over an extended period of time, it is not clear to me why a monsoon would not fit in with the scaling relationships derived in this paper. Would that not be an end-member condition for considering the role of water infiltration in setting the spatial distribution of landslides on lower slopes? If not, then why not? I imagine other people not as familiar with monsoon dynamics like myself might ask the same question.

Line 14: Does the proportion of flat ground affect the slide aspect ratios as well, since the flat ground may provide more accommodation space for runout?

Line 22: This sentence cannot be true, as Figs. 6 and 7b all show a relationship between storm metrics on landslide scar areas. Do you mean other storm metrics outside of storm total rainfall?

Line 20: This is almost certainly true, especially for the smaller-area landslides that depend on local slope smaller than what a 30 m pixel can resolve.

Technical Corrections

Line 3: "..we have very few datasets of rainfall-induced landslides." I think this should be clarified that this is the case only for single-event inventories.

Line 6: should be "orders of magnitude"

Line 8: "The non-linear scaling with total rainfall." Two notes: 1) "nonlinear" is one word; 2) the variable that is being scaled with total rainfall should be specified.

Line 11: "contrarily" should be "contrary"

Line 18: "..itself expected to increase with global change." I find this sentence slightly confusing. Consider replacing "itself" with "which are"

Line 26: should be "This progress has been possible"

Line 5: Should be "is needed"

Line 7: Sentence fragment ". . .affected by the storm, and thus . "

Lines 13-18: Nice overview!

Line 20: should be "datasets" (plural)

Line 25: should be "The rainfall was"

Line 30: "details" should be singular

Sections 2.1 and 2.2 – Introduce acronyms (e.g., B08) as you introduce each dataset.

Line 29: "(30m)" I think a space usually goes between the value and the units, e.g., "30 m". This should be done consistently throughout the manuscript.

Figure 1. It would be very helpful to make the landslide inventory polygons and rain

gauge stations contrast more with the background. The topography could easily be represented as a hillshade since the absolute elevations are not the focus of the figure. Also, the panels are not labeled with letters as described in the caption.

In the caption, "Landslides inventory" should be "Landslide inventories"

Line 2: "sub-parts" could be "subsets" perhaps?

Lines 2-3: The values and units appear italicized in one instance and standard font in the other. Should be consistently reported.

Lines 5-6: I think "rain gauge" is the correct spelling

Lines 13-14: Is the correlation between D and Rt and I3 and Rt shown anywhere? May be good for the Supplemental.

Line 21: Probably good to put a reference here where the increased runout for larger volume slides is discussed (e.g., Legros, 2002).

Line 8: "However, for subset with less" should be "However, for subsets with fewer"

Line 10: "uncertainties" should be "uncertainty"

Line 1: Sentence could be simplified here, e.g., "Landslide inventories typically exhibit heavy-tailed, power law frequency size distributions"

Line 14: "Fitted" should be "Fit"

Line 9: the term "a slope gradient units" is confusing.

Line 10: before the word "oversampling", "and" should be "an"

Line 30: "create" should be "creates"

Line 34: Should there be a figure reference here?

Line 7: Should reference Fig. 5d here.

Figure 5: In Figure 5b, you could define the axis value more clearly on the axis itself. "Landslide sampling on steep slopes" does paint a clear image of what the axis value represents.

Unit labels are italicized and inconsistent with other unit labels throughout the text.

Line 4: remove "S??"

Line 3: Could reference Fig 5b specifically.

References

Densmore, A. L. & Hovius, N. Topographic fingerprints of bedrock landslides. Geology 28, 371–374 (2000). Keefer, D. K. The importance of earthquake-induced landslides to long-term slope erosion and slope-failure hazards in seismically active regions. Geomorphology 10, 265–284 (1994). Legros, F. The mobility of long-runout landslides. Eng. Geol. 63, 301–331 (2002). Reid, M. E. & Iverson, R. M. Gravity-Driven Groundwater Flow and Slope Failure Potential. Water Resour. Res. 28, 939–950 (1992).

**ESurfD**

---

## Author Comment (AC2) · 23 May 2018

NOTE : In the following documents referees comments are in normal fonts and the answer to them are in bold fonts.

Review of Marc et al. "Towards a global database of rainfall-induced landslide inventories: first insights from past and new events" by David Milledge.

Major Comments

This is a well executed study with novel and interesting findings. I have three general comments and a large number of minor comments but neither the major nor minor comments reflect a fundamental problem in the research in my view.

I am not convinced that it is essential (or helpful) to present your inventories as the only inventories that are suitable for this type of analysis (as you seem to do on P2-3). Instead you could simply say they are one set of inventories and they demonstrate the power of this type of approach. I am not convinced of the need for landslides beneath an entire storm footprint to be mapped and am sceptical that entire storm footprints can be convincingly defined so I'm not convinced by your critique of studies that analyse far smaller study areas (other than on sample size grounds).

**>> The mostly agree with the referee. We will develop the description along these lines:**

**1/ number is always important, in absolute term because below 50-100 landslide the reliability of any statistical treatment is uncertain but also in relative term, because a statistical study based on 500 landslides out of a storm that caused ~5000 take the risk to have biased interpretation if the (potential) specificity of the subset population are not notcied and understood (e.g., mostly large landslide ?, mostly landslide near river ? Mostly landslide in a given lithology ? Etc etc ).**

**2/ If enough landslides are mapped within an AOI (e.g., >50-100 in total, > 75% of landslide above the resolution limit, leading to a reasonable frequency-size distribution ) the inventory above the AOI is likely to be statistically usable and representative of the various processes and conditions affecting the process in the AOI. Then a partial inventory will indeed allow to study any local parameter and their variations within the AOI : e.g. landslide density, landslide size distribution, relations to slope etc.**

**3/ However, a comprehensive inventory may have the additionaly advantage to gather enough landslide across different areas (in terms of lithology, relief etc) potentially allowing to establish a hierarchy in controlling parameters and also allowing to study an averaged landslide response less likely to be dominated by specific site effects. comprehensive inventories are the only ones allowing to study the variations of total landsliding.**

**We will include a synthetic description of these point in the early part of the manuscript. These should answer the various minor comments about the importance of "comprehensive" landslide catalog, and better acknowledge the potential use of "partial" inventories.**

The methodology description could be more consistent between inventories. Similar information is reported for each case but the style of the reporting differs and some key information reported in some cases is not present in others (e.g. image source, image resolution, acquisition date).

**>> We will work on a higher degree of consistency.**

**However, we disagree on key image source/resolution/date not beeing present, as all this information is in Suppl Table 1. Is the referee suggesting that we integrate this table within the main text ?**

**Or simply specifying some terms in the main text ?**

I am not convinced that your focus on 'comprehensive' inventories is necessary nor that examination of total landslide numbers, volumes or areas are particularly meaningful in relation to rainfall triggered landslide inventories (though I think the findings on landslide density and slope are extremely interesting and thought provoking). This focus might reflect a desire for comparability to co-seismic landslides but I think the two triggers are importantly different. For example, it is extremely difficult to define the spatial and temporal limits on a single storm. In addition I find the results relating to total numbers, volumes and areas less convincing because they are predicted from a small number of point rainfall records. A clearer explanation of why 'comprehensive' inventories and total statistics are important would be a valuable addition to the paper.

Minor Comments

P2
L30: comprehensive: this term needs defining.
**>> We mean that all landslides detectable above the resolution limit were mapped, and that we could observe the landslide density fading away in all direction, indicating the limits of the footprint of the high intensity part of the storm.**
**We will specify this two criterium in the main text.**

P3
L7: comprehensive mapping: where do you start and finish. Your definition of a storm is very important here and I don't see it at the moment. For example shouldn't the Morakot mapping extend to the Phillipenes and China on this basis?
**>> The pragmatic answer is that the landslide response in Philippines and China was negligible compared to the one in Taiwan (We doubled check that quickly by looking at Landsat images where no hyperpicnal flows or alluviations in stream exiting hilly areas are visible, contrarily to Taiwan where these processes are very clear). Same is true if we look at landsliding in the rest of Japan hit by Typhoon Talas progressing Northward after hitting the Kii peninsula. This may be in part due to topographic difference, but I think this is also due to the fact enormous amount of rainfall was poured on this topography, probably largely because of orographic effects (cf Chien and Kuoa 2009, Taniguchi et al., 2009). Thus preceding rainfall on less high topography (e.g. in the Philippines) probably received much less rainfall, and the following rainfall over China (or Japan for Talas) was also likely less simply because little or no recharge ove the ocean was possible and a significant fraction of the typhoon moisutre was used up.**
**A theoretical answer is more difficult to find and would require a proper definition of a storm event. A tentative meteorological definition could consider a mass of moving moisture with a single source of moisture. A typhoon or (afternoon) convective cell could be such an object, that can then travel and pour its accumulated moisture as rainfall (and or snowfall) over an area limited in space an time. However, for the landslide community, only the relatively high intensity part of this rainfall matters, (as the part being below the landslide threshold can be neglected) and the spatial and temporal limits of a storm event could be further limited ( as in the case of Morakot and Talas, where the orographic rainfall effect limit greatly the part of the typhoon relevant for landsliding).**

**Chien, F.-C. and Kuo, H.-C.: On the extreme rainfall of Typhoon Morakot (2009), J. Geophys. Res., 116(D5), D05104, doi:10.1029/2010JD015092, 2011.**

**Taniguchi, A., Shige, S., Yamamoto, M. K., Mega, T., Kida, S., Kubota, T., Kachi, M., Ushio, T. and Aonashi, K.: Improvement of High-Resolution Satellite Rainfall Product for Typhoon Morakot (2009) over Taiwan, J. Hydrometeor., 14(6), 1859–1871, doi:10.1175/JHM-D-13-047.1, 2013.**

L30: this gets at a difficult issue, what do you include as a landslide? I think you need a clear definition that can be applied across all inventories and I don't see one at present. Divergence from the definition in different inventories will introduce bias to your results.
**>> We tried to avoid mapping (or remove in the already mapped inventories (J11, TW9) deposition and erosion in the fluvial system, broadly defined as the areas with permanent flows, visible in the high resolution image. This meant that debris flow on hillslope would be mapped but not its prolongation within the fluvial system. We considered bank collapses as a disturbance that would be localized, usually not symmetric and not necessary linked to a landslide/debris flow on the hillsllopes. Clearly we may miss some bank collapse, and where to put the limit between a debris flow on a hillslope and its continuation on the fluvial channel is difficult and somewhat subjective. However, if amalgamation is avoided, the width estimate (and thus scar area and volume) will be relatively insensitive to these issues, that mainly affect the total runout and aspect ratio.**

P7
L4: fluvial system: How do you define the fluvial system and how did you identify it for the study area?
**>>The fluvial system was broadly defined as the areas with permanent flows, visible in the high resolution image. As we do not perform any anaylsis on the relation between the hydrographic network and landslide this appraoch is just aiming at making sure landslides are limited to hillslopes (cf comment P3 L30).**

L20-21: maximal forcing: this doesn't seem to be consistent with your argument for the importance of complete landslide footprints. You are comparing the forcing at a single location within the footprint to the properties of the entire footprint.
**>> Well the maximum forcing is taken as a "storm magnitude", and it is compared to total landsliding and peak landslide density close of this maxima l forcing, so we think the approach is reasonable.**
**We think that the issue of the referee is that in the introduction we push for "comprehensive landslide inventory" and later do not make fully use of it.**

**This is for 2 reasons: 1/ We have access to extensive rainfall data constraining the spatial pattern of rainfall for only a few cases, and the analysis of the spatial pattern is beyond the scope of this study and left for a future study (Marc et al., in Prep).**
**2/ Generally a comparison of storm magnitude with total landsliding requires an accurate order of magnitude of the total landsliding. We think that the fact that storm magnitude correlates well with total landslidng suggest some internal correlation between the peak total rainfall, and the mean rainfall and its variability within the storm footprint. Still we acknowledge that such correlation may not hold for all type of rainfall events: In our database, small events are likely**

brief convective thunderstorm (C99,C15), while large ones are typhoons (M02,TW8,TW9, J11), which are very large singular system that are loaded during their displacement over ocean, and unloaded on landfall, even more importantly when hitting high relief. They fit well in a tentative definition of a storm event based on a single process / source of moisture accumulation and subsequent downpour on a given spatial/temporal zone (cf comments P3 L7).

  The cases of B08 and B11 are more complex as they may results from interactions between multiple oceanically sourced moisture and specific meteorologic conditions on land. As mentionned in the main text, other rainfall period such as a monsoon could also rather be characterized as the sum of repeated convection  events then transported across India, not allowing to differentiate individual meteorological event.

        We will try to add some elements of this discussion into section 2.3

P9
L5: how, and where, did you measure landslide width?
--> The width was initially measured by GIS on a limited number (~50) of randomly selected slides in Colombia and Japan. To make this point more robust we proceded as follow:
Text added in the revised manuscript:
"We measured systematically the width of 419 randomly selected landslides across all range of polygon area and aspect ratio, in the following inventories : J11, TW8, B11, and C15. The width was measured on the upper part of the landslide only, the likely scar, and ~4 width measurements made in arcGIS were averaged. When compared to the equivalent width obtained through our runout correction, 72% of the polygons are within 30% of the measured width and 96% within a factor of 2 (Fig Suppl X). We do not observed a trend in bias with area nor aspect ratio, except perhaps for the automatically mapped landslide in B11, where high aspect ratio correlates with underestimated width"

[Figure]

**Figure Suppl X : Ratio of Estimated to measured width against landslide area (left) and landslide equivalent aspect ratio (right) for 418 landslides randomly selected during 4 rainfall events ( TW8, B11, J11, C15). Most of estimated width are within 30% of the measured width, as shown by the horizontal black lines.**

L6: I think you could state this more simply by saying that you assume that scars have equal length and width. This is the same assumption used by Pelletier et al., 1997.
**--> This would have been an option. However, we became aware of a study presented at EGU general assembly of 2018, where the aspect ratio of a number of landslide scar has been analyzed. Domej et al., 2017, reported that in average the length-width ratio remained close of 1.5 for all landslide size. So we follow them and assume that 1.5 represent a good average of the length width ratio of landslide scar.**

**References:**
**Domej, G., Bourdeau, C. and Lenti, L.: Mean Landslide Geometries Inferred from a Global Database of Earthquake- and Non-Earthquake-Triggered Landslides, Italian Journal of Engineering Geology and Environment, (2), 87–107, doi:10.4408/IJEGE.2017-02.O-05, 2017.**

P10
L24: to what extent is the landslide distribution area constrained by your study area (i.e. the extent of available images). Taking this to an extreme did Typhoon Morakot trigger landslides in China or the Phillipenes and should these also be included? This again reflects something that I think you need to discuss somewhere, the differences between rain storms and earthquakes as triggers: where are they similar enough to borrow frameworks from one another and where do they differ?

**>> Well it is clear that, contrarily to earthquake that have a well delimited source (across the fault), the rainfall forcing is moving together with the storm and can travel other significant areas.**

P11
L24: peculiar distributions: are these distributions peculiar if you are seeking power laws but not if other alternatives are considered? Have you tried a log-normal distribution? Negative curvature of the tail in log-log space sometimes indicates better fits for log-normal distributions?
**>> The question is difficult to solve and not so important for our studies: Fitting log-normal distribution by MLE we obtain better agreement for some distributions and worst for others (comparing the Kolmogorov-Smirnov Test statistic and Anderson Darling test statistic obtained for log-normal or IGD fit obtained by MLE).**
**In other work the Inverse Gamma distributions has been found to provide the best fit to 3 large landslide catalogue (Malamud 2004). Further some work on the theoretical emergence of landslide size distribution also predict power-law decaying tail (Stark and Guzzetti 2009), with a tail related to the mechanical properties of the medium, implying the debate may not only be a question of goodness of fit, especially given that some datasets may be affected by artifacts.**

**Although we can mention these facts, solving such a debate is clearly out of the scope of our work. We will of course double check wether or not LogNormal fit parameters (Mu and Sigma) are correlated to rainfall parameters.**

[Figure]

**Figure for the discussion: Comparison of lognormal distribution (Solid) and Inverse gamma distribution (dashed) for the best fit of landslide scar size distribution. Fit are obtained by MLE.**

L29: aspect ratio below 2: why below 2? What are the specifics of the equation? I had understood it to be A=w^2, which would give an aspect ratio of 1.
**--> This is now updated based on Domej 2017 (Cf comments above). This is a reduction to an**

**aspect ratio of 1.5.**

L11: focussing on scar areas seems sensible but this particular approach seems strange and the choice of modal topographic slope somewhat arbitrary, could you provide a more robust explanation for this choice? Alternatively couldn't you have used your previously defined scar area (w^2) to identify scars as the highest w^2 area of each polygon? This would be consistent with your previous definition and would avoid introducing an arbitrary slope threshold which could bias the results.

**--> We note that for C99 there is not much difference between initiation point and the steeper part of the landslides (i.e, after the mode). Additionally we simply do not interpret what happens below the topogaphic mode but can certainly show it. Reducing all landslides polygon to their scar would require some work but is possible and we will try to examine whether or not it creates any difference in the results.**

**P13**

L14: Could you use line thickness to indicate the slope beyond which small numbers of cells in the value range preclude interpretation of the line? It would be useful for the reader to know where that point is for each dataset. Also could you colour the lines in Fig 5 by storm duration? This might make it easier to pick out the behaviour you are identifying in the text and to make a connection between 5A and 5B.

**>> Actually we plan to use the notion of prediction interval of the landscape slope distribution to assess if the landslide-affected slope distribution is staistically different from the landscape one.**

**With this method we will indicate which part of the distribution can be robustly interpreted and which one are less robust.**

**We will change the colour-code to reflect the duration, this is a good suggestion.**

P17L9: Total storm rainfall: These results are extremely interesting. They suggest that absolute rainfall properties are good predictors for landslide properties. In the rainfall threshold literature there has been debate over whether absolute rainfall properties are driving failure or whether it is the degree of deviation from normal conditions (e.g. expressed as percentiles). It might be useful if you could reflect on this in relation to your findings. Would a plot of rainfall percentiles for these storms look very similar to the plot of absolutes that we see here?

**>> By deviation of from normal conditions, do the referee means the comparison between the sotm rainfall and for example mean annual rainfall or mean seasonnal rainfall ? Or more something like the estimated return period of such a storm ? It may be difficult to estimate one or the other for a number of events but we can try (Mean annual or seasonnal rainfall may be tractable).**

**It is clear that although total rainfall may be a good predictor of the relative amount of landsliding between different storm ( as shown in Fig 6 and 7) the control on landsliding must be more compliacted as in the surrounding area similar rainfall occur without triggering landslides (Taiwan, Japan), or in the same season similar total rainfall did not trigger landslide (in Colorado), so either antecedent rainfall or some constraints on intensity will be needed to generalize /strengthen the results we found.**

**In any case we will add such caveat somewhere in the discussion.**

P18L17: we have no clear physical explanation: isn't this something that either extreme rainfall community or the hurricane community have thought about? It would be useful to point readers to

key reference from that literature here even if you don't strongly back one particular explanation.
**>> We will look at this literature to try to suggest interesting reads. Some work indeed study the correlation between the total rainfall on land of hurricane and tropical storm with their diameter and travel velocity (Jiang, et al., 2008). We will try to relate to such work.**

**Jiang, H., Halverson, J. B., Simpson, J. and Zipser, E. J.: Hurricane "Rainfall Potential" Derived from Satellite Observations Aids Overland Rainfall Prediction, J. Appl. Meteor. Climatol., 47(4), 944–959, doi:10.1175/2007JAMC1619.1, 2008.**

References
Clauset, A., Shalizi, C.R. and Newman, M.E., 2009. Power-law distributions in empirical data. SIAM review, 51(4), pp.661-703.
Cox, N.J., 2007. Kernel estimation as a basic tool for geomorphological data analysis. Earth surface processes and landforms, 32(12), pp.1902-1912.
White, E.P., Enquist, B.J. and Green, J.L., 2008. On estimating the exponent of power-law frequency distributions. Ecology, 89(4), pp.905-912.

Anonymous Referee #2

General Comments **( Summary paragraphs NOT reproduced here)**

In terms of the general aspects of the manuscript presentation and layout, I find that the paper needs some fine-tuning and clarification, but overall it is close to being a finished product. The abstract provides a clear summary of the work, and the overall structure and segmentation of the manuscript is easy to follow. The title largely makes sense, although I find the last section stating "first insights from past and new events" confusing since all datasets are within the past twenty years and the youngest event occurred in 2015, and I'm not sure what that phrase adds to the description of the research. There are a number of places where language needs to be altered slightly, and I've tried to provide examples below in the technical comments section of the review. Mathematical formulae appear to be largely correct, but abbreviations for the landslide inventories (although intuitive) are not defined before they are used. Additionally, I found that Figure 2 should be modified, as it is very difficult to see the pink landslide polygons draped over the red and green topography. I imagine this would be especially difficult for people who are red-green colorblind. The supplementary material complements the manuscript well, and I have a few comments regarding supplementary figures below. Although I have a few additional concerns related to content and clarification, overall I think this paper will make an interesting contribution to Esurf..

**>> We thank the reviewer for its interest in our study and findings.**

**From these general comments we retain :**
**1/ A clarification of Figure 1 (not 2) showing the landslide inventories.**

**2/ Some edits of the title : We could indeed drop the second statement.**
**An interesting alternative, slightly more descriptive could be "Towards a global database of rainfall-induced landslide inventories: first insights on landscape scale landsliding caused by rainfall event"**
**This option includes somewhat the notion of global magnitude of landsliding ( e.g., Fig 6,7) and spatial distribution witin the landscape ( e.g., Fig 5).**
**I would welcome comments of Referees and AE on such a title, and if they oppose it, we could simply stop with "Towards a global database of rainfall-induced landslide inventories", although it does not leave a hint than the paper do not only report on collating data but also analyze and interpret them.**

**3/ Check that all landslide variables are defined in the text, and improve and correct texts, following both Referees technical comments.**

1st Paragraph: I'm not sure I agree with the statement that the goal of constraining quantitative relationships between landslide occurrence and rainfall is out of reach. The authors cite examples of this in the same paragraph. I do agree that there is certainly room for improvement in his area, which I think is the implied sentiment here.
**>>Ok, we will rephrase in this direction.**

Equation 1. I appreciate that the authors' goal here is to try to bridge the gap between purely deterministic models and purely statistical models, but I think that there needs to be a little more clarity. At the end of Paragraph 1, for example, you state that certain parameters such as permeability and cohesion that are required for deterministic approaches make a landscape-scale approach in data-poor regions inapplicable, yet you specifically include those parameters in your idealized semi-deterministic Equation 1. Why then is a deterministic approach not appropriate? I think that a bit more discussion might clarify these discrepancies.
**>> Deterministic approach will require a fine scale representation of porosity and its variability at a fine scale: more or less the one of the landslides, so at 10-100m. In a semi-deterministic approach we may need only a constrain on the mean porosity (and perhaps some other aspect of its distribution like its variance or skewness) within a whole catchment or 10x10km catchment. Obtaining such information remains a challenge but may be more tractable, and may be correlated to other large scale observable (From hydrological behavior to soil maps ?).**

Lines 31-34: I'm confused by this sentence. When specifically are data from 2010-2012 used? When May-June 2009 data are not available for a specific location in the landslide-affected area?
**>> Exactly. We will clarify this, but in most places in this case imagery just after the event is available in Google Earth.**

Section 2.2 overall.
I'm also confused with the general methodology here. You map landslides on 30 m Landsat imagery, as well as on higher resolution imagery within Google Earth, but only in areas where a negative change in NDVI was observed at the 30 m scale. You then say that field mapping in the area reports twice as many landslides than was observed via remote sensing, but that the missing landslides must be smaller than #1 m resolution. Could the missing landslides not just be in areas that didn't result in a

negative NDVI shift in the landsat imagery? For example, a small translation or slump in a forested area may not affect a 30 m pixel.

**>> This is true, it would however be relatively small landslide: indeed landslide much smaller than a landsat pixel ( e-g, 10x10m rather than 30x30m) that looked fresh were almost systematically causing reduction of NDVI in one or 2 pixel, although the NDVI reduction was smaller than for large landslides.**
**So the Landsat NDVI is very sensitive to sub-pixel size landsliding, but the Google Earth imagery is essential to only map landslides and not many other anthropogenic/biological processes changing the NDVI.**

Lines 23-27: In your discussion of peculiar landslide frequency distributions, you focus on deviations from the (perhaps) expected Inverse-Gamma distributions at the large end of the distributions. What about deviations on the smaller end? For example, in the Total Area distributions, TW9, B11, C99, and J11 deviation from the maximum likelihood estimations pretty substantially for small landslide areas. Converting total landslides area to landslide scar area (As), the TW9 distribution especially deviates quite far from the expected P values. Is there a known reason for these deviations? I am far from an expert on landslide frequency distributions, but it seems worth discussing since it is quite apparent on Figure 3!

**>> In agreement with Referee 1 we will also add a few line about the quality of IGD vs LogNormal (or essentially exponential tail). However, this will remain superficial given deciding on the functional form of size distribution is clearly out of the scope of our paper.**

**For the deviation in small landslide size, they are difficult to interpret because after the roll-over, censoring issues and difficulty to distinguish multiple adjacent landslides are likely important even with high resolution imagery. Given that these deviations are important almost only in TW9 where amalgamation and mapping artifacts were frequent but could not be mitigated down to the smaller landslide sizes, I would not try to interpret them.**

Figure 3: Similarly, I don't believe it is mentioned why the authors choose to break up their landslide populations into two groups. Is this just to more easily visualize? Or is it based on the quality of datasets?

**>> Indeed the split is just for visualization. We will specify it in the caption.**

Lines 11-12: Is there a plot that shows the relationship described on these lines? I couldn't find one. Maybe it would be worth including these in the supplemental material.

**>> Ok we will add such a figure  in the supplement.**

Lines 8-9: Very cool.

**>> Thank you, this is indeed exciting.**

Lines 12-15: If there is a continuous forcing of heavy rainfall over an extended period of time, it is not clear to me why a monsoon would not fit in with the scaling relationships

derived in this paper. Would that not be an end-member condition for considering the role of water infiltration in setting the spatial distribution of landslides on lower slopes? If not, then why not? I imagine other people not as familiar with monsoon dynamics like myself might ask the same question.

**>> The reviewer is certainly right about the slope sitribution, but not for the scaling between total rainfall and landsliding. To clarify both point we added:**

**Page 18 L20: "Indeed, in a long period with fluctuating rainfall such as the monsoon, drainage and storage of water will certainly not be negligible and the derivation of a soil water content proxy will be necessary (cf., Gabet et al., 2004) ."**

**P20 L23 we will add "A testable hypothesis would be that large populations of landslides caused exclusively during the monsoon would exhibit a strong undersampling of the steepest slopes"**

Line 14: Does the proportion of flat ground affect the slide aspect ratios as well, since the flat ground may provide more accommodation space for runout?

**>> Maybe, but these sentence really relates to the availability of topography within the footprint of the storm event, so the landscape scale rather than the landslide scale, where local slope variations can indeed influence runout.**

Line 22: This sentence cannot be true, as Figs. 6 and 7b all show a relationship between storm metrics on landslide scar areas. Do you mean other storm metrics outside of storm total rainfall?

**>> Here, the reviewer confuses the total area (counting only scar or whole landslide) shown in Fig 6A, 6B and 7, and the individual scar area distribution.shown in Figure 3.**

**To avoid such confusion for other readers we rephrase to :**

**"We do not find a clear influence of storm metrics on the probability distribution of individual landslide scar areas or landslide runout (Fig 3)."**

Line 20: This is almost certainly true, especially for the smaller-area landslides that depend on local slope smaller than what a 30 m pixel can resolve.

**>> We also think that. Thus an analysis with a high resolution DEM maybe needed.**

---

## Referee Comment (RC3) · D. Milledge (Referee) · 24 May 2018

Marc et al. provide full and useful responses to my initial comments including good suggestions on how they will modify the manuscript to address these comments. Thank you for posting these here before the end of the discussion period and giving me an opportunity to respond.

One comment in particular that you sought reviewers' views on was the title of the paper so I will paste that discussion here.

R2 commented that: The title largely makes sense, although I find the last section

stating "first insights from past and new events" confusing since all datasets are within the past twenty years and the youngest event occurred in 2015, and I'm not sure what that phrase adds to the description of the research.

The authors responded that: We could indeed drop the second statement. An interesting alternative, slightly more descriptive could be "Towards a global database of rainfall-induced landslide inventories: first insights on landscape scale landsliding caused by rainfall event" This option includes somewhat the notion of global magnitude of landsliding ( e.g., Fig 6,7) and spatial distribution witin the landscape ( e.g., Fig 5). I would welcome comments of Referees and AE on such a title, and if they oppose it, we could simply stop with "Towards a global database of rainfall-induced landslide inventories", although it does not leave a hint than the paper do not only report on collating data but also analyze and interpret them.

My view is that: I would prefer something that indicates your key findings, which to me are: the breakdown in local slope dependence and that landslide density depends on storm total rainfall. Perhaps: "Initial insights from a global database of rainfall-induced landslide inventories: the weak influence of slope and strong influence of total storm rainfall" These are fairly strong claims but I think they reflect your findings. Alternatively I am fine with: "Towards a global database of rainfall-induced landslide inventories: insights on landscape scale landsliding caused by rainfall events" I removed first because it is a little ambiguous whether you mean initial insights from these events (which would be fine) or the first ever insights on landscape scale landsliding (which would not).

David Milledge

Please also note the supplement to this comment:
https://www.earth-surf-dynam-discuss.net/esurf-2018-20/esurf-2018-20-RC3-supplement.pdf

―――――――――――――――――――――

**ESurfD**

Interactive
comment

**Supplement:**

NOTE : In the following documents referees comments are in normal fonts and the answer to them are in bold fonts. My response to the authors' answers are in normal fonts.

Review of Marc et al. "Towards a global database of rainfall-induced landslide inventories: first insights from past and new events" by David Milledge.

Major Comments

This is a well executed study with novel and interesting findings. I have three general comments and a large number of minor comments but neither the major nor minor comments reflect a fundamental problem in the research in my view.

I am not convinced that it is essential (or helpful) to present your inventories as the only inventories that are suitable for this type of analysis (as you seem to do on P2-3). Instead you could simply say they are one set of inventories and they demonstrate the power of this type of approach. I am not convinced of the need for landslides beneath an entire storm footprint to be mapped and am sceptical that entire storm footprints can be convincingly defined so I'm not convinced by your critique of studies that analyse far smaller study areas (other than on sample size grounds).

**>> The mostly agree with the referee. We will develop the description along these lines:**

**1/ number is always important, in absolute term because below 50-100 landslide the reliability of any statistical treatment is uncertain but also in relative term, because a statistical study based on 500 landslides out of a storm that caused ~5000 take the risk to have biased interpretation if the (potential) specificity of the subset population are not notcied and understood (e.g., mostly large landslide ?, mostly landslide near river ? Mostly landslide in a given lithology ? Etc etc).**

**2/ If enough landslides are mapped within an AOI (e.g., >50-100 in total, > 75% of landslide above the resolution limit, leading to a reasonable frequency-size distribution ) the inventory above the AOI is likely to be statistically usable and representative of the various processes and conditions affecting the process in the AOI. Then a partial inventory will indeed allow to study any local parameter and their variations within the AOI : e.g. landslide density, landslide size distribution, relations to slope etc.**

**3/ However, a comprehensive inventory may have the additionaly advantage to gather enough landslide across different areas (in terms of lithology, relief etc) potentially allowing to establish a hierarchy in controlling parameters and also allowing to study an averaged landslide response less likely to be dominated by specific site effects. comprehensive inventories are the only ones allowing to study the variations of total landsliding.**

**We will include a synthetic description of these point in the early part of the manuscript. These should answer the various minor comments about the importance of "comprehensive" landslide catalog, and better acknowledge the potential use of "partial" inventories.**

I think this will address my concern here. I still think your points boil down to: arguments of sample size, which I agree with (point 1 & 3); and systematic sampling, i.e. all identifiable landslides larger than some minimum size, which I agree with (point 1 &2). You don't demonstrate here that multiple smaller inventories that do not cover the 'full storm footprint' are less valid than a smaller number of 'complete' inventories. It seems to me that sample size is the key here. That is fine, your sample is very large and that makes it a particularly useful contribution, I think building an argument round this rather than 'event completeness' would be more compelling.

The methodology description could be more consistent between inventories. Similar information is reported for each case but the style of the reporting differs and some key information reported in some cases is not present in others (e.g. image source, image resolution, acquisition date).

**>> We will work on a higher degree of consistency.**

**However, we disagree on key image source/resolution/date not beeing present, as all this information is in Suppl Table 1. Is the referee suggesting that we integrate this table within the main text ? Or simply specifying some terms in the main text ?**

The table is useful but it would strengthen the paper to make the reporting more consistent within the methods text. This could be done by using consistent language / terms (and covering similar material) for each inventory as you suggest.

I am not convinced that your focus on 'comprehensive' inventories is necessary nor that examination of total landslide numbers, volumes or areas are particularly meaningful in relation to rainfall triggered landslide inventories (though I think the findings on landslide density and slope are extremely interesting and thought provoking). This focus might reflect a desire for comparability to co-seismic landslides but I think the two triggers are importantly different. For example, it is extremely difficult to define the spatial and temporal limits on a single storm. In addition I find the results relating to total numbers, volumes and areas less convincing because they are predicted from a small number of point rainfall records. A clearer explanation of why 'comprehensive' inventories and total statistics are important would be a valuable addition to the paper.

This comment didn't receive a specific response but it overlaps with other discussion on how to define "rainstorm event landslide inventories" in which I think we've reached broad consensus.

Minor Comments

P2 L30: comprehensive: this term needs defining.

**>> We mean that all landslides detectable above the resolution limit were mapped, and that we could observe the landslide density fading away in all direction, indicating the limits of the footprint of the high intensity part of the storm. We will specify this two criterium in the main text.**

This is addressed.

P3 L7: comprehensive mapping: where do you start and finish. Your definition of a storm is very important here and I don't see it at the moment. For example shouldn't the Morakot mapping extend to the Phillipenes and China on this basis?

**>> The pragmatic answer is that the landslide response in Philippines and China was negligible compared to the one in Taiwan (We doubled check that quickly by looking at Landsat images where no hyperpicnal flows or alluviations in stream exiting hilly areas are visible, contrarily to Taiwan where these processes are very clear). Same is true if we look at landsliding in the rest of Japan hit by Typhoon Talas progressing Northward after hitting the Kii peninsula. This may be in part due to topographic difference, but I think this is also due to the fact enormous amount of rainfall was poured on this topography, probably largely because of orographic effects (cf Chien and Kuoa 2009, Taniguchi et al., 2009). Thus preceding rainfall on less high topography (e.g. in the Philippines) probably received much less rainfall, and the following rainfall over China (or Japan for Talas) was also likely less simply because little or no recharge ove the ocean was possible and a significant fraction of the typhoon moisutre was used up.**

**A theoretical answer is more difficult to find and would require a proper definition of a storm event. A tentative meteorological definition could consider a mass of moving moisture with a single source of moisture. A typhoon or (afternoon) convective cell could be such an object, that can then travel and pour its accumulated moisture as rainfall (and or snowfall) over an area limited in space an time. However, for the landslide community, only the relatively high intensity part of this rainfall matters, (as the part being below the landslide threshold can be**

neglected) and the spatial and temporal limits of a storm event could be further limited ( as in the case of Morakot and Talas, where the orographic rainfall effect limit greatly the part of the typhoon relevant for landsliding).

**Chien, F.-C. and Kuo, H.-C.: On the extreme rainfall of Typhoon Morakot (2009), J. Geophys. Res., 116(D5), D05104, doi:10.1029/2010JD015092, 2011.**

I think summarising this discussion within the text would be a good way forward so that it is clear what your working definition of a storm (and a storm footprint) is and what complexities might be neglected in this.

L30: this gets at a difficult issue, what do you include as a landslide? I think you need a clear definition that can be applied across all inventories and I don't see one at present. Divergence from the definition in different inventories will introduce bias to your results.

**>> We tried to avoid mapping (or remove in the already mapped inventories (J11, TW9) deposition and erosion in the fluvial system, broadly defined as the areas with permanent flows, visible in the high resolution image. This meant that debris flow on hillslope would be mapped but not its prolongation within the fluvial system. We considered bank collapses as a disturbance that would be localized, usually not symmetric and not necessary linked to a landslide/debris flow on the hillsllopes. Clearly we may miss some bank collapse, and where to put the limit between a debris flow on a hillslope and its continuation on the fluvial channel is difficult and somewhat subjective. However, if amalgamation is avoided, the width estimate (and thus scar area and volume) will be relatively insensitive to these issues, that mainly affect the total runout and aspect ratio.**

This explanation is useful and goes some way to (and even beyond) addressing my comment. I think it would be useful to include in the text something along the lines: "Here we define a landslide as…" it might include something that captures your distinction above that fluvial transport and bank collapse are not included in this definition though the precise boundary between these and a landslide may be fuzzy. It would also be useful to highlight that these are rapid landslides – you don't include landslides that do not remove the covering vegetation I guess so you won't see earth flow activity (and probably wouldn't want to include it anyway since you would then need to talk about velocity changes rather than presence or absence).

L20-21: maximal forcing: this doesn't seem to be consistent with your argument for the importance of complete landslide footprints. You are comparing the forcing at a single location within the footprint to the properties of the entire footprint.

**>> Well the maximum forcing is taken as a "storm magnitude", and it is compared to total landsliding and peak landslide density close of this maxima l forcing, so we think the approach is reasonable. We think that the issue of the referee is that in the introduction we push for "comprehensive landslide inventory" and later do not make fully use of it.**

**This is for 2 reasons: 1/ We have access to extensive rainfall data constraining the spatial pattern of rainfall for only a few cases, and the analysis of the spatial pattern is beyond the scope of this study and left for a future study (Marc et al., in Prep).**

**2/ Generally a comparison of storm magnitude with total landsliding requires an accurate order of magnitude of the total landsliding. We think that the fact that storm magnitude correlates well with total landslidng suggest some internal correlation between the peak total rainfall, and the mean rainfall and its variability within the storm footprint. Still we acknowledge that such correlation may not hold for all type of rainfall events: In our database, small events are likely brief convective thunderstorm (C99,C15), while large ones are typhoons (M02,TW8,TW9, J11), which are very large singular system that are loaded during their displacement over ocean, and unloaded on landfall, even more importantly when hitting high relief. They fit well in a tentative definition of a storm event based on a single process / source of moisture accumulation and subsequent downpour on a given spatial/temporal zone (cf comments P3 L7).**

The cases of B08 and B11 are more complex as they may results from interactions between multiple oceanically sourced moisture and specific meteorologic conditions on land. As mentionned in the main text, other rainfall period such as a monsoon could also rather be characterized as the sum of repeated convection events then transported across India, not allowing to differentiate individual meteorological event.

**We will try to add some elements of this discussion into section 2.3**

This would be a useful addition to the paper and addresses my comment.
* * *
P9 L5: how, and where, did you measure landslide width?

**--> The width was initially measured by GIS on a limited number (~50) of randomly selected slides in Colombia and Japan. To make this point more robust we proceded as follow: Text added in the revised manuscript:**

**"We measured systematically the width of 419 randomly selected landslides across all range of polygon area and aspect ratio, in the following inventories : J11, TW8, B11, and C15. The width was measured on the upper part of the landslide only, the likely scar, and ~4 width measurements made in arcGIS were averaged. When compared to the equivalent width obtained through our runout correction, 72% of the polygons are within 30% of the measured width and 96% within a factor of 2 (Fig Suppl X). We do not observed a trend in bias with area nor aspect ratio, except perhaps for the automatically mapped landslide in B11, where high aspect ratio correlates with underestimated width"**

I had misunderstood your area correction method. You don't actually calculate scar area based on measured width and assumed aspect ratio. You calculate scar area based on mapped area and perimeter by estimating width assuming landslides are elliptical then using an assumed scar aspect ratio to translate to area. The width test figure that you include is useful and gives some sense of error in width estimates. Perhaps you could also clarify your method in the manuscript.
* * *
P10 L24: to what extent is the landslide distribution area constrained by your study area (i.e. the extent of available images). Taking this to an extreme did Typhoon Morakot trigger landslides in China or the Phillipenes and should these also be included? This again reflects something that I think you need to discuss somewhere, the differences between rain storms and earthquakes as triggers: where are they similar enough to borrow frameworks from one another and where do they differ?

**>> Well it is clear that, contrarily to earthquake that have a well delimited source (across the fault), the rainfall forcing is moving together with the storm and can travel other significant areas.**

Agreed, and discussed elsewhere in this review-response document.
* * *
P11 L24: peculiar distributions: are these distributions peculiar if you are seeking power laws but not if other alternatives are considered? Have you tried a log-normal distribution? Negative curvature of the tail in log-log space sometimes indicates better fits for log-normal distributions?

**>> The question is difficult to solve and not so important for our studies: Fitting log-normal distribution by MLE we obtain better agreement for some distributions and worst for others (comparing the Kolmogorov-Smirnov Test statistic and Anderson Darling test statistic obtained for log-normal or IGD fit obtained by MLE). In other work the Inverse Gamma distributions has been found to provide the best fit to 3 large landslide catalogue (Malamud 2004). Further some work on the theoretical emergence of landslide size distribution also predict power-law decaying tail (Stark and Guzzetti 2009), with a tail related to the mechanical properties of the medium, implying the debate may not only be a question of goodness of fit, especially given that some datasets may be affected by artifacts. Although we can mention these facts, solving such a debate is clearly out of the scope of our work. We will of course double check wether or not LogNormal fit parameters (Mu and Sigma) are correlated to rainfall**

**parameters. Figure for the discussion: Comparison of lognormal distribution (Solid) and Inverse gamma distribution (dashed) for the best fit of landslide scar size distribution. Fit are obtained by MLE.**

I agree that this is difficult to solve and not central to this study. I think noting that the distributions are not always a particularly good fit to IGD but in some cases may be better fit by LogNormal would probably be enough. I think theoretical work explaining power law tails is a distraction since that work explicitly set out to explain power law tails.
* * *
L29: aspect ratio below 2: why below 2? What are the specifics of the equation? I had understood it to be A=w^2, which would give an aspect ratio of 1.

 **--> This is now updated based on Domej 2017 (Cf comments above). This is a reduction to an aspect ratio of 1.5.**

This issue is resolved.
* * *
L11: focussing on scar areas seems sensible but this particular approach seems strange and the choice of modal topographic slope somewhat arbitrary, could you provide a more robust explanation for this choice? Alternatively couldn't you have used your previously defined scar area (w^2) to identify scars as the highest w^2 area of each polygon? This would be consistent with your previous definition and would avoid introducing an arbitrary slope threshold which could bias the results.

**--> We note that for C99 there is not much difference between initiation point and the steeper part of the landslides (i.e, after the mode). Additionally we simply do not interpret what happens below the topogaphic mode but can certainly show it. Reducing all landslides polygon to their scar would require some work but is possible and we will try to examine whether or not it creates any difference in the results.**

This would be a good addition.
* * *
P13 L14: Could you use line thickness to indicate the slope beyond which small numbers of cells in the value range preclude interpretation of the line? It would be useful for the reader to know where that point is for each dataset. Also could you colour the lines in Fig 5 by storm duration? This might make it easier to pick out the behaviour you are identifying in the text and to make a connection between 5A and 5B.

**>> Actually we plan to use the notion of prediction interval of the landscape slope distribution to assess if the landslide-affected slope distribution is staistically different from the landscape one. With this method we will indicate which part of the distribution can be robustly interpreted and which one are less robust. We will change the colour-code to reflect the duration, this is a good suggestion.**

This sounds good.
* * *
P17L9: Total storm rainfall: These results are extremely interesting. They suggest that absolute rainfall properties are good predictors for landslide properties. In the rainfall threshold literature there has been debate over whether absolute rainfall properties are driving failure or whether it is the degree of deviation from normal conditions (e.g. expressed as percentiles). It might be useful if you could reflect on this in relation to your findings. Would a plot of rainfall percentiles for these storms look very similar to the plot of absolutes that we see here?

**>> By deviation of from normal conditions, do the referee means the comparison between the sotm rainfall and for example mean annual rainfall or mean seasonnal rainfall ? Or more something like the estimated return period of such a storm?**

I was thinking largely of percentile based approaches, which use somewhat more information than a straight deviation from the mean but are generally less complete than a return period.

**It may be difficult to estimate one or the other for a number of events but we can try (Mean annual or seasonnal rainfall may be tractable). It is clear that although total rainfall may be a good predictor of the relative amount of landsliding between different storm ( as shown in Fig 6 and 7) the control on landsliding must be more compliacted as in the surrounding area similar rainfall occur without triggering landslides (Taiwan, Japan), or in the same season similar total rainfall did not trigger landslide (in Colorado), so either antecedent rainfall or some constraints on intensity will be needed to generalize /strengthen the results we found. In any case we will add such caveat somewhere in the discussion.**

Adding something like this would be useful and would address my comment. The other analysis that you suggest would be extremely interesting but may not be possible here and that would be fine by me.
* * *
P18L17: we have no clear physical explanation: isn't this something that either extreme rainfall community or the hurricane community have thought about? It would be useful to point readers to key reference from that literature here even if you don't strongly back one particular explanation.

**>> We will look at this literature to try to suggest interesting reads. Some work indeed study the correlation between the total rainfall on land of hurricane and tropical storm with their diameter and travel velocity (Jiang, et al., 2008). We will try to relate to such work. Jiang, H., Halverson, J. B., Simpson, J. and Zipser, E. J.: Hurricane "Rainfall Potential" Derived from Satellite Observations Aids Overland Rainfall Prediction, J. Appl. Meteor. Climatol., 47(4), 944–959, doi:10.1175/2007JAMC1619.1, 2008.**

This sounds good.
* * *
From R2 comments – where the authors seek reviewers' opinions:

The title largely makes sense, although I find the last section stating "first insights from past and new events" confusing since all datasets are within the past twenty years and the youngest event occurred in 2015, and I'm not sure what that phrase adds to the description of the research.

**2/ Some edits of the title : We could indeed drop the second statement. An interesting alternative, slightly more descriptive could be "Towards a global database of rainfall-induced landslide inventories: first insights on landscape scale landsliding caused by rainfall event" This option includes somewhat the notion of global magnitude of landsliding ( e.g., Fig 6,7) and spatial distribution witin the landscape ( e.g., Fig 5). I would welcome comments of Referees and AE on such a title, and if they oppose it, we could simply stop with "Towards a global database of rainfall-induced landslide inventories", although it does not leave a hint than the paper do not only report on collating data but also analyze and interpret them.**

I would prefer something that indicates your key findings, which to me are: the breakdown in local slope dependence and that landslide density depends on storm total rainfall.  Perhaps:

"Initial insights from a global database of rainfall-induced landslide inventories: the weak influence of slope and strong influence of total storm rainfall"

These are fairly strong claims but I think they reflect your findings. Alternatively I am fine with:

"Towards a global database of rainfall-induced landslide inventories: insights on landscape scale landsliding caused by rainfall events"

I removed first because it is a little ambiguous whether you mean initial insights from these events (which would be fine) or the first ever insights on landscape scale landsliding (which would not).

---

## Author Comment (AC3) · 6 Jul 2018

**Dear Editor,**

**Please find below the detailed answer to both referees major and minor comments. These corrections have been implemented in the text, several supplementary figures have been added (width estimate validation, correlation of the storm parameters, correlation between landsliding and climate normalized rainfall , ... ), and several main text figures have been updated.**

**We almost never opposed the referees comments, and performed some re-analyses as they suggested. Two modifications arise from these (re)-analyses :**
**1/ The correlation between storm duration and landslide position and slope faded, and is now only evocated. The result that rainfall induced landslides do not strongly oversample steep slopes remain and is characterized more robustly.**
**2/ A correlation between landslide scar areas and storm metrics was found and it is now presented and discussed.**

**The rest of our results and discussion presented before has not significantly changed and we hope is now clearer.**

Review of Marc et al. "Towards a global database of rainfall-induced landslide inventories:
first insights from past and new events" by David Milledge.
Major Comments
This is a well executed study with novel and interesting findings. I have three general comments and
a large number of minor comments but neither the major nor minor comments reflect a
fundamental problem in the research in my view.

I am not convinced that it is essential (or helpful) to present your inventories as the only inventories
that are suitable for this type of analysis (as you seem to do on P2-3). Instead you could simply say
they are one set of inventories and they demonstrate the power of this type of approach. I am not
convinced of the need for landslides beneath an entire storm footprint to be mapped and am
sceptical that entire storm footprints can be convincingly defined so I'm not convinced by your
critique of studies that analyse far smaller study areas (other than on sample size grounds).

The methodology description could be more consistent between inventories. Similar information is
reported for each case but the style of the reporting differs and some key information reported in
some cases is not present in others (e.g. image source, image resolution, acquisition date).
**>> To clarify the methodology, we first include a short paragraph about the general methodology. Then we have made sure to include all imagery type, dates and spatial resolutions for each event.**
**The style remains somewhat different given the variations in methodology/specific knowledge for each case.**

I am not convinced that your focus on 'comprehensive' inventories is necessary nor that examination
of total landslide numbers, volumes or areas are particularly meaningful in relation to rainfall
triggered landslide inventories (though I think the findings on landslide density and slope are
extremely interesting and thought provoking). This focus might reflect a desire for comparability to
co-seismic landslides but I think the two triggers are importantly different. For example, it is
extremely difficult to define the spatial and temporal limits on a single storm. In addition I find the
results relating to total numbers, volumes and areas less convincing because they are predicted from
a small number of point rainfall records. A clearer explanation of why 'comprehensive' inventories
and total statistics are important would be a valuable addition to the paper.

Minor Comments
P1L4: Associated to: How do you know that these events are associated to one another.
**>>We replaced "associated to" with "coincident with" : all the inventory presented can be dated within a few hours or days leaving a single rain-storm as a trigger. Of course this does not exhaust the question of antecedent "preparatory" conditions.**

P2
L8: deterministic approaches inapplicable: I think this statement is a little strong. Is it really fair to

say that they are inapplicable given their data requirements.

**>> Original meaning was that in those places without enough data (i.e. most places) the deterministic approach is inapplicable. We rephrase to:**
**"In most places, such level of detailed information is currently unavailable, rendering deterministic approaches hardly applicable."**

L30: comprehensive: this term needs defining.

**>> We mean that all landslides detectable above the spatial resolution limit were mapped, and that we could observe the landslide density fading away in all direction, indicating the limits of the footprint of the high intensity part of the storm.**
**We will specify this two criteria in the main text. We added in the introduction :**
**"By comprehensive inventories we mean that all landslides larger than a given size were mapped, and that the extent of the imagery allowed to observe the landslide density fading away in all direction, tracking the reduction of the forcing intensity of the triggering event, whether shaking or rainfall."**

L34: Why is it insufficient? I think you need to demonstrate this. Is this a sample size argument? Some things won't be possible to calculate but others will. What can you and can't you do with a subset inventory and how big does the subset need to be?. Is it ever possible to capture the full inventory for a storm? How do you define its bounds?

**>> Following the referee advice we focus this point on sample size and biased sampling. We also refer to fragmentary inventories rather than catchment scale :**
**We rephrased to : "Although relatively rare, some case studies based on fragmentary event inentories exist (and are briefly reviewed in the next section) but they may contain too few landslides for statistical analyses or may be biased to specific locations (along roads or near settlements, within weak lithological units, downslope, etc), thus complicating the deconvolution of forcing versus site influences."**

**We address the question of storm definition and bound elsewhere (cf P3 L7).**

P3

L4: landslide scale: I am not clear what this means. Could you define it?

**>>We meant studies that focus on small samples of individual landslides, and landslide scale was ambiguous there. We now rephrase it:**
**"... we consider that neither studies based on small sample of individual landslide or on global-scale analysis will be able to constrain effectively Eq 1..."**

L7: comprehensive mapping: where do you start and finish. Your definition of a storm is very important here and I don't see it at the moment. For example shouldn't the Morakot mapping extend to the Phillipenes and China on this basis?

**>> The pragmatic answer is that the landslide response in Philippines and China was negligible compared to the one in Taiwan (We doubled check that quickly by looking at Landsat images where no hyperpycnal flows or alluviations in stream exiting hilly areas are visible, contrarily to Taiwan where these processes are very clear). Same is true if we look at landsliding in the rest of Japan hit by Typhoon Talas progressing Northward after hitting the Kii peninsula. This may be in part due to topographic difference, but I think this is also due to the fact enormous amount of rainfall was poured on this topography, probably largely because of orographic effects (cf Chien and Kuo 2009, Taniguchi et al., 2009). Thus preceding rainfall on less high topography (e.g. in the Philippines) probably received much less rainfall, and the following rainfall over China (or Japan for Talas) was also likely less heavy because little or no recharge over the ocean was possible and a significant fraction of the typhoon moisture was used up.**

**More generally a proper "object" definition is hardly achievable in Meteorology as we deal with a continuous fluid which suffers perturbations with scales interactions and as meteorological events are not independent from each other. Therefore not all atmospheric specialists agree on the definition and limit of a single 'storm' . Ideally, future studies could categorize storms according to some space-time filtering and analyze their relations with landslides for each storm category. Currently, our database in not sufficient for this.**

**Also our ability to measure the characteristic of a 'storm' or rainfall event depends on its spatio-temporal extension (sampling a large and long lasting storm is more robust than sampling a very localized event)**

**We added in the discussion, P19 L30: "In any case, several caveat should be taken with the preliminary scaling between total storm rainfall and total landsliding. First the definition and limit of a single "storm" is not generally**

agreed in the meteorological community, because the atmospheric fluids suffers perturbations with scale interactions, and therefore with events not independent from each other. Ideally, future studies could categorize storms according to some space-time filtering and analyze the scaling with total landsliding for each storm category. Currently, our data base is not sufficient for this yet."

L14: adequately: how do you quantify adequate representation, what would inadequate representation look like and how do you know whether a representation is adequate?
>> We meant an adequate representation of the landslide population, i.e., one population that represent (and thus average) the different processes that lead to landsliding. This means a sufficient number, with comprehensive representation of the different landslide size above a threshold etc. This point is developed in comments P4L4.

We replaced "representing adequately" by a neutral term : "containing".

L30: this gets at a difficult issue, what do you include as a landslide? I think you need a clear definition that can be applied across all inventories and I don't see one at present. Divergence from the definition in different inventories will introduce bias to your results.
>> We tried to avoid mapping (or remove in the already mapped inventories (J11, TW9) deposition and erosion in the fluvial system, broadly defined as the areas with permanent flows, visible in the high resolution image. This meant that debris flow on hillslope would be mapped but not its prolongation within the fluvial system. We considered bank collapses as a disturbance that would be localized, usually not symmetric and not necessary linked to a landslide/debris flow on the hillslopes. Clearly we may miss some bank collapse, and where to put the limit between a debris flow on a hillslope and its continuation on the fluvial channel is difficult and somewhat subjective. However, if amalgamation is avoided, the width estimate (and thus scar area and volume) will be relatively insensitive to these issues, that mainly affect the total runout and aspect ratio.
We added at the start of the section 2.2:
"Here we consider landslides as a rapid downslope transport of material, disturbing vegetation outside of the fluvial domain, which we define by visible water flow in the imagery. We also consider individual landslides with a single source or scar areas to avoid amalgamation, and split polygons when necessary. Although the transition between hillslopes and channel may be blurry and in part subjective, the width estimation (cf. 2.4) will mitigate variations of the transport length, as long as large alluviated, or flooded areas are not mapped as landslide deposits. "

P4L4: whether or not the statistical properties of a subset are representative: you need to demonstrate that they are not representative for your argument here to hold and it is not obvious that this is the case. They might not be representative because of the sample size but why should you need the all landslides triggered by a particular storm, it seems reasonable to assume that one catchment is independent of another for these processes and on these timescales.
>> We mostly agree with the referee. We will develop the description along these lines:

1/ number is always important, in absolute terms because below 50-100 landslides the reliability of any statistical processing is uncertain but also in relative terms, because a statistical study based on 500 landslides out of a storm that caused ~5000 take the risk to have biased interpretation if the (potential) specificity of the subset population are not noticed and understood (e.g., mostly large landslide ?, mostly landslide near river ? Mostly landslide in a given lithology ? Etc etc ).
2/ If enough landslides are mapped within an AOI (e.g., >50-100 in total, > 75% of landslides above the spatial resolution limit, leading to a reasonable frequency-size distribution ) the inventory above the AOI is likely to be statistically usable and representative of the various processes and conditions affecting the process in the AOI. Then a partial inventory will indeed allow to study any local parameter and their variations within the AOI : e.g. landslide density, landslide size distribution, relations to slope etc.
3/ However, a comprehensive inventory may have the additional advantage to gather enough landslide across different areas (in terms of lithology, relief etc) potentially allowing to establish a hierarchy in controlling parameters and also allowing to study an averaged landslide response less likely to be dominated by specific site effects. comprehensive inventories are the only ones allowing to study the variations of total landsliding.

We removed this sentence and the next and replaced it by :

"These inventories could not constrain the total landslide response to a storm, but may allow to constrain

**relationships between landslide properties and local rainfall, provided that enough landslides have been mapped for statistical analysis (e.g., >50-100) and without any systematic sampling bias. However, a detailed assessment of these datasets properties and of their relation to rainfall is out of the scope of this study although it would probably complement interestingly our work in the future."**

**We also rephrased a conclusion sentence: "… show the value of mapping systematically a large sample of the landslides that can be related to a single storm ..."**

P5
L15-19: why take this approach rather than breaking up the multi-headed polygons manually?
**--> In this study, the multiheaded polygons have often equivalent width and it is likely that they contribute equally to the overall runout and deposit. Breaking up the different head to reflect equivalent contribution would have been impractical.**
**We rephrased : "The surface of the source areas were often of similar width, suggesting equivalent contribution from each source to the transport and deposit areas, and rendering a manual splitting impractical. "**

L6-17: this methods description is difficult to follow. Image acquisition dates and image resolution information is missing in some cases. It would also be useful to give some indication of the performance of the automated classification with respect to manually mapped landslides.
>> We specified the dates and resolution of the Geoeye and Landsat 7 images.
**"30m Landsat 7 images from February 2011 "**

**"very high resolution Geoeye-1 image (2/0.5 m resolution in multispectral/panchromatic) from the 26th of May 2010 and the 20th of January 2011."**

P7
L1: there is quite a long window between pre and post event imagery in some cases. How confident can you be that another storm did not trigger some of the landslides? What evidence do you have that this is the case?
**>> Although this may have not been clear enough, instantaneous imagery was available in M02, C99,  B11 (90%), J11 and most of C15 (~85%).  The 2 years were probably referring to C15 where  we have few cloud less images on the AOI, but the event is also dated by a HR images in Google Earth from 31st of May 2015 (cdf P7 L12 comment below). The other intervals are much shorter and without other storms of the same magnitude.**

**USGS reports with VHR image also shows ~50-75% of landslides in C15. B08 (90%zone)**
**M02 and C99 were verified fresh on the field. For the other case we rely on pre-event imagery :**
**Pre-event Google Earth imagery in B11 in most place.**
**15 July for J11.**
**For TW 2009, we have a clear Landsat image on 24 of June (day 175, for some reason it was noted 255 in TableSuppl 1) and then the Formosat images were taken on the week after Morakot. Thus although some landslide may have happen during smaller scale storms before the start of August it would be negligible compared to the number of slides triggered by Morakot.**
**For TW8 we have 32 days between the mapping images. In the south the intermediate image is foggy but allows to see intact vegetation on the hillslope. Only the Northernmost landslide cluster, near Sun Moon Lake is fully masked by clouds but some Aster images allow to confirm that most of what is mapped was not present before mid July. Also, the rainfall from the 15th to 20th of July was much stronger than anything in the previous 3 weeks.**

L2: mapped automatically: I think you need to include a methods description for this automatic mapping and some information on how the quality of this mapping was evaluated.
**>>  We had made some confusion here, as the landslides were not automatically mapped there. Only the mapping quality was problematic as many slides were connected through channel deposits and alluviation.  We rephrased : "For subsets of the inventory, especially to the East of the main divide, landslides were significantly amalgamated and bundled with river channel alluviation. We thus manually split the polygons and removed the channel areas."**

L4: fluvial system: How do you define the fluvial system and how did you identify it for the study

area?
>>**The fluvial system was broadly defined as the areas with permanent flows, visible in the high resolution image. As we do not perform any analysis on the relation between the hydrographic network and landslide this approach is just aiming at making sure landslides are limited to hillslopes. Therefore we added in our definition of landslide : "the fluvial domain, which we define by visible water flow in the imagery." (cf comment P3 L30)**

L5: to map at least the largest: I don't understand what this means, were there some areas of your
study area that you did not have high resolution imagery for? If so what fraction of the study area
was this and what impact does this have on the inventory as a whole?
**--> Yes in Taiwan 2008, and 2009, or in small sub-area of Brazil 2008, or Japan 2011, we could not get high resolution image close enough in time to confidently validate our low resolution mapping done (in TW8 and B11) or the emergency mapping of NILIM done with airphotos in Japan.**
**In figure 1, the yellow boxes shows footprint of VHR images (Google Earth in most cases) sufficiently close in time. In TW8 this is the whole inventory. IN Taiwan 2009, the pan-sharpened Formosat imagery has 2m spatial resolution, and only the cloud covered areas (~5%) where mapped with low spatial resolution Landsat images. In B08 and B11 <10%, in J11 ~15%.**

**We rephrased : "In a few areas with clouds (~300 km² in total or <5% of the AOI) in the post-event mosaic, we mapped with Landsat 5 images (from 24th of June and 12th of September 2009), even if the spatial resolution limit may have censored the smallest landslides."**

**We also specified the area covered by HR imagery :**
**In Japan : "With Google Earth we could validate NILIM on about 85\% of the AOI and we added ..."**
**In Brasil 2008: "Therefore we used extensively high-resolution imagery available in Google Earth ( over >90\% of the AOI) acquired in May-June 2009"**

**In Brasil 2011: "we mapped landslides directly from Google Earth (available over >90\% of the AOI)"**

**The impact of low spatial resolution is mainly on the landslide number, but not likely significant on the total area and volume or even landslide density, as discussed in the results section.**

L12: Specific dates are missing for the Landsat images. This is a 2 year window, which seems a very
long time. How confident can you be in assigning landslides to a single event within that window and
what is the basis for this confidence? This is particularly important given your earlier critique of
other inventories
>> **We forgot to indicate it here but Google Earth contains a mostly cloud free images covering most of the dataset dated from 31st May 2015. The scar look very fresh in this image and corresponds well with the "new" scars observed by comparing the 2016-2014 images.**
**We specify now:**
**"Landslide mapping was carried out by comparing a (10 m) Sentinel 2 image from the 21st of July 2016 and a pan-sharpened (15 m) Landsat 8 image from the 19th of July and the 26th December of 2014. These images were selected for their absence of clouds, good conditions of light and similarity. High spatial resolution imagery from Google Earth, dated from the 31st of May 2015 shows fresh scars consistent with our mapping over most of the area (Fig 1), and we assumed that the remaining landslides (<15% of the inventory) were also triggered by the same rainfall event. "**

L20-21: maximal forcing: this doesn't seem to be consistent with your argument for the importance
of complete landslide footprints. You are comparing the forcing at a single location within the
footprint to the properties of the entire footprint.
>> **Well the maximum forcing is taken as a "storm magnitude", and it is compared to total landsliding and peak landslide density close of this maxima l forcing, so we think the approach is reasonable.**
**We think that the issue of the referee is that in the introduction we push for "comprehensive landslide inventory" and later do not make fully use of it. This for 2 reasons: 1/ We have access to extensive rainfall data constraining the spatial pattern of rainfall for only a few cases, and the analysis of the spatial pattern is beyond the scope of this study and left for a future study (Marc et al., in Prep).**

**2/** Generally a comparison of storm magnitude with total landsliding requires an accurate order of magnitude of the total landsliding. We think that the fact that storm magnitude correlates well with total landsliding suggest some internal correlation between the peak total rainfall, and the mean rainfall and its variability within the storm footprint. Still we acknowledge that such correlation may not hold for all type of rainfall events: In our database, small events are likely brief convective thunderstorm (C99,C15), while large ones are typhoons (M02,TW8,TW9, J11), so very large singular system that are loaded during their displacement over ocean, and unloaded on landfall, even more importantly when hitting high relief. The cases of B08 and B11 is more complex as it more likely results from interactions between oceanic moisture and specific meteorologic conditions on land.

We specify now : "… but rather maximal forcing, which may be taken as a storm magnitude, ..."
and at the start of the next paragraph:
"A more detailed analysis of the spatio-temporal pattern of the rainfall and of its relations to the spatial pattern of landsliding is highly desirable, but challenging and is left for a future study."

In the discussion, after mentionning the diffulty to define a storm we added :
"Second, linking total rainfall in a limited area and the total landsliding within the storm footprint implicitly suggests that storm rainfall is somewhat structured with some internal correlations between peak rainfall, storm size, and the spatial pattern of rainfall intensity within the storm. This seems to be the case for large tropical storms (Jiang_ et al.,2008), but should be explored for a broader range of storm types. Orographic effects (e.g., Houze, 2012; Taniguchi et al., 2013), focussing high intensity rainfall on topographic barriers, may also enhance such correlation between local total rainfall, and the broader pattern of rainfall and landsliding."

L30: landslide densities: calculated over what window size, I think that this choice will be critically important. On a small window density will have multiple local peaks.
--> We forgot to specify it in the main text ( but it is on the caption of Fig S2) : 0.05x0.05° so approximately 25km². It is true that multiple peak can appear at smaller resolution, however considering only the magnitude of the peak landslide density, the trend observed in Fig X was not changing for cells of 0.02° to 0.1°.

L33: why not use 3 gauges for Colorado? Where were the next nearest gauges and why were they discounted?
>> The second closest gauge (LB) is ~2 km North and 3km West from the Grizzly Peak station, at the limit of the airphoto area in Fig 2, and it recorded 13mm on the 28[th] of July (43 for Grizzly Peak). The third stations (JG) is a couple of kilometer south of the study area and recorded 20 mm. Radar data shown by Godt and Coe (2007) indicates the afternoon rainfall reached 25-35mm/hr along the peaks eastward and northeastward of the GP stations while it was only ~6mm/hr near LB or at the southern edge of the study area (presumably to JG).
We will add in the main text:
"For the Colorado 1999 storm, radar data indicate very localized high intensity precipitation located on the peaks were debris flow occurred (Godt and Coe 2007) and suggest that the single closest gage is more representative than averaging with the other nearby ones."

P8L11: continuous period: I'm not totally clear what this means, does it mean that if there was no rain in a 3 hour period then that is the end / start of the storm? Was the same duration criteria applied to all records?
>> Yes we meant that the conditions I3<3 mm/hr was the start/end of the storm. However, this can be directly apply only in TW8,TW9, J11, C99, M02 we do not have this information for B08 B11 and the threshold for the satellite data in C15 may be different, as mentioned in the text.
Thus we just added "continuous period when rainfall was sustained (I3>3mm.hr-1)"

P9
L5: how, and where, did you measure landslide width?
--> The width was initially measured by GIS on a limited number (~50) of randomly selected slides in Colombia and Japan. To make this point more robust we proceeded as follow:
We rephrased in the revised manuscript (also updating our assumption about the aspect ratio):
"To validate this geometric method to retrieve landslide width we measured systematically the width of 418 randomly selected landslides across all range of polygon area and aspect ratio, belonging to four inventories: J11, TW8, B11, and C15. For each polygon, we focused on the upper part of the landslide only, the likely scar, and averaged 4 width (i.e., length perpendicular to flow) measurements made in arcGIS. The width estimated based on P

and A are within 30% and 50% of the measured width for 72% and 92% of the polygons, respectively (Fig Suppl 5).We do not observed a trend in bias with area nor aspect ratio, except perhaps for the automatically mapped landslide in B11, where high aspect ratio correlates with underestimated width. Thus, for correctly mapped polygons we can use P and A to derive W and a proxy of landslide scar area, $A\_s \sim 1.5 W^2$. We assume landslide scars have an aspect ratio of 1.5, as it was found to be the mean aspect ratio found across a range of landslide size within a global database of 277 measured landslide geometries (Domej et al., 2017). Even if this equivalent scar area may not exactly correspond to the real landslide scar, it effectively removes the contribution of the landslide runout to the landslide size and allows to compare different size distributions while reducing the impact of variable runout distances."

This should also clarify that we compute W based on Perimeter and Area.

L6: I think you could state this more simply by saying that you assume that scars have equal length and width. This is the same assumption used by Pelletier et al., 1997.
--> This would have been an option. However, we became aware of a study presented at EGU general assembly of 2018, where the aspect ratio of a number of landslide scar has been analyzed. Domej et al., 2017, reported that in average the length-width ratio remained close of 1.5 for all landslide size. So we follow them and assume that 1.5 represent a good average of the length width ratio of landslide scar.

Cf Previous Comment P9 L5

P10
L23: isolated remote landslides: how were these defined?
>> We refer to single or small group of landslides without other landslides in the surrounding, usually on the fringes of the landslide population.

We specify now in the text : "(i.e., single or small cluster of landslides without other landslides within 5-10 km)"

L24: to what extent is the landslide distribution area constrained by your study area (i.e. the extent of available images). Taking this to an extreme did Typhoon Morakot trigger landslides in China or the Phillipenes and should these also be included? This again reflects something that I think you need to discuss somewhere, the differences between rain storms and earthquakes as triggers: where are they similar enough to borrow frameworks from one another and where do they differ?

>> Well it is clear that, contrarily to earthquake that have a well delimited source (across the fault), the rainfall forcing is moving together with the storm and can travel over significant areas.

This aspect has been discussed in comments P3 L7 and P7 L20-21

P11
L3: typically have power law: they have typically been fit with these distributions but do we know that they typically follow that distribution or do we fit power laws tailed distributions without testing alternatives (e.g. log-normal).
>> Some studies have tested various distributions option and find Pareto distribution (Stark and Hovius 2001, 1 dataset) or Inverse Gamma distributions to fit best (Malamud 2004, 3 datasets) but this debate is out of the scope of our work.
We rephrase with : "have typically been fit by power-law tailed distributions."

L19: must also: must is a strong statement, could it alternatively be due to different mapping criteria?
-> We changed "physical parameters must influence" to "However, mechanical parameters are also expected to influence the roll-over position (Stark and Guzzetti, 2009, Frattini and Crosta 2013), as suggested by the fact that MI2, mapped with 1m resolution ... "
 as these studies have propose physical models ( of slope stability or probabilistic rupture propagation) where both the roll-over and power-law decay are influenced by mechanical properties.

L24: peculiar distributions: are these distributions peculiar if you are seeking power laws but not if other alternatives are considered? Have you tried a log-normal distribution? Negative curvature of the tail in log-log space sometimes indicates better fits for log-normal distributions?

**>> The question is difficult to solve and not so important for our studies: Fitting log-normal distribution by MLE we obtain better agreement for some distributions and worst for others (comparing the Kolmogorov-Smirnov Test statistic and Anderson Darling test statistic obtained for log-normal or IGD fit obtained by MLE).**

**In other work the Inverse Gamma distributions has been found to provide the best fit to 3 large landslide catalogue (Malamud 2004). Further some work on the theoretical emergence of landslide size distribution also predict power-law decaying tail (Stark and Guzzetti 2009), with a tail related to the mechanical properties of the medium, implying the debate may not only be a question of goodness of fit, especially given that some datasets may be affected by artifacts.**

**Although we can mention these facts, solving such a debate is clearly out of the scope of our work. Thus we simply added the following statement : "In these two cases, for whole landslide area or landslide scar only, we note that a MLE fit of a lognormal distribution agrees better to the data (based on the result of both the Kolmogorov-Smirnov and the Anderson-Darling test). In other inventories a lognormal fit is equivalent or worse than an IGD, but the functional form of landslide size distribution and its implication are beyond the scope of this study. "**

L26: Why use a least square fit to represent the power law tail? The problems associated with using least squares fits to binned data rather than an MLE have been widely discussed (e.g. White et al., 2008; Clauset et al., 2009) and Clauset et al. (2009) provide appropriate tools to fit only the power law tail using an MLE.
**--> We removed the least square fit and only present the IGD and LogNormal MLE estimates. We updated the text consequently (cf previous comment).**

**Updating all exponents we found that although whole landslide area is not correlated to rainfall a correlation exist for landslide scar area, with larger rainfall causing more large landslides, thus we updated the result section and added a main text figure. :**

**"The decay exponents of the distribution of landslide whole area do not correlate significantly with any storm metrics (Intensity, duration or total rainfall) ($|R|<0.1$). However, after runout normalization, the decay exponent of landslide scar area correlate with all metrics, although with significant scatter ($R^2 \sim 0.5$ Fig \ref{area}, Suppl. 7). The two largest storms (J11 and TW9) have the lowest exponents ($\alpha+1\sim1.8$), and thus a large proportion of very large landslides, while the two smallest storms (C15 and C99) have a small proportion of large landslides and large exponents ($\alpha+1\sim2.7$). However, intermediate cases are very scattered, as B11 and TW8 have similar total rainfall, peak intensity and duration but very different distribution with $\alpha+1=1.9$ and with $\alpha+1=1.9$ , respectively."**

**And in section 3.2.3:**
**" The decay exponents of the distribution of landslide whole area do not correlate significantly with any storm metrics (Intensity, duration or total rainfall) (|R|<0.1). However, after runout normalization, the decay exponent of landslide scar area correlate with all metrics, although with significant scatter (R^2 ~ 0.5 Fig 4, 7). The two largest storms (J11 and TW9) have the lowest exponents (alpha+1 ~1.8), and thus a large proportion of very large landslides, while the two smallest storms (C15 and C99) have a small proportion of large landslides and large exponents (alpha+1 ~2.7). However, intermediate cases are very scattered, as B11 and TW8 have similar total rainfall, peak intensity and duration but very different distribution with alpha+1 =1.9 and with alpha+1 =2.6 , respectively. Still, randomly removing one event (i.e., jackknife sampling) we obtained $R^2$ between 0.4 and 0.7, with a similar mean $R^2$ about 0.5. "**

**The discussion, conclusion and abstract have also been updated to mention this positive influence (even if partial) on the size distribution.**

L29: aspect ratio below 2: why below 2? What are the specifics of the equation? I had understood it

to be A=w^2, which would give an aspect ratio of 1.

**--> This is now updated based on Domej 2017 (Cf comments above). This is a reduction to an aspect ratio of 1.5. We corrected that in the main text.**

P12
L4-8: Why is this censoring of low slopes necessary? I am not clear on what you are trying to achieve by removing them?

**>> The valley floors have a slope distribution mainly between 0 and 10°, likely strongly affected by DEM uncertainties. Still in some zone, most of the affected area may be dominated by floodplain and hillslopes may be a fraction of the AOI. But landslide scar never occur in the valley plain and should be compared to hillslopes rather than to these floodplains. However, a slope criteria cannot be used (as there may be slope <5° within the hillslope domain (ridges, shoulder). Instead we use an elevation criteria for Micronesian Island, and we masked out the flat areas in B08. We added: "allowing to obtain a hillslope distribution as an approximate Gaussian, with a mode significantly beyond our threshold of 5°."**

L8-10: generating a histogram then smoothing it seems an unusual approach to this problem, results will likely be sensitive to both the smoothing window and smoothing function. Given the theoretical basis for Kernel density estimation (e.g. Cox, 2007), why not use this approach?

**>> We thank the referee for his suggestion. We reanalyze the slope data with a kernel density smoothing, using a normal kernel and optimized bandwidth. We rephrased :**
**"To focus on the scar area of each landslide polygon, we extracted only the slopes for the highest elevation pixels representing a surface of 1.5*W.^2. Then, we computed the probability density function for the landslide affected area and whole topography (hereafter the "landslide" and "topographic" distribution) with a normal-kernel smoothing with an optimized bandwidth, as implemented in Matlab."**

P13
L5: initiation point: I don't think you have previously defined this or explained how these points are identified.

**--> We added in the data section on past inventories: "Initiation point were assumed to be the highest point upslope of each mapped landslides. In 57 out of 328 polygons multiple initiation points (2 to >15) were mapped for multi-headed polygons (Godt and Coe 2007)."**

L11: focussing on scar areas seems sensible but this particular approach seems strange and the choice of modal topographic slope somewhat arbitrary, could you provide a more robust explanation for this choice? Alternatively couldn't you have used your previously defined scar area (w^2) to identify scars as the highest w^2 area of each polygon? This would be consistent with your previous definition and would avoid introducing an arbitrary slope threshold which could bias the results.

**--> We followed the referee suggestion and we extracted the most elevated fraction of each polygon corresponding to an area of ~1.5*W². See P12 L8-10 comment.**

**Doing this, all curves have slightly shifted to higher slopes. But the main results that most rainfall events do not tend to trigger landslides on very steep slopes remain. Most events do not over-sample more than 1.5-2 times the topographic distribution, much less than the ChiChi example or the C99 rainfall event.**

L14. Could you use line thickness to indicate the slope beyond which small numbers of cells in the value range preclude interpretation of the line? It would be useful for the reader to know where that point is for each dataset. Also could you colour the lines in Fig 5 by storm duration? This might make it easier to pick out the behaviour you are identifying in the text and to make a connection between 5A and 5B.

**>> We now computed the prediction interval for a random draw out of the topographic distribution, and only interpret the landslide probability that significantly differs from it.**

**We added : "An important issue is to determine whether the landslide probability can be considered a random drawing from slopes of the topography or not. Given that landsliding affect less than 10\% of the landscape, the sampling of the topography by landslides can be approximated by a Bernoulli sampling. In this case, the central limit theorem gives the 95\% prediction interval as P_T +/- 1.96 \sqrt(P_T(1-P_T)/N), with N the number of independent**

draws, here taken as the number of landslide scars. The convergence of N draws to P_T within the prediction interval is only valid if N>30, N P_T>5 and $N (1-P_T)>5$, implying that only very large samples can be interpreted towards the extremity of the topographic slope distribution, where P_T is small."

And in the figure caption: "The ratios are estimated with the PDF averaged within 3° bins. Solid circles and dots represent ratios where the landslide probability is beyond or within, respectively, the 95\% prediction interval of the topography distribution. Crosses indicate bins where data are insufficient for the validity of the Central Limit Theorem required to estimate prediction interval."

Importantly, the case of C15 with the smallest landslide number (N=171) does not allow to distinguish confidently topographic and landslide distribution at high slopes. Thus only C99 significantly oversample topographic slopes. We updated the results and the discussion ass follows:

Results : in 3.1.2: "For all events we observe that $P_L$ is significantly different from a random drawing of the topography with oversampling of the slopes beyond $S_M$ and undersampling below it (Fig 4B). However, we note that for most events, the undersampling and oversampling is smaller than a factor of 2. Some cases (C15, J11 and TW8) have stronger oversampling ($>4$) for $S-S_M >25$ but they may not be representative ratios given the limited number of landslides and of slopes thus steep (i.e. $N P_T <5$). The scars of C99 clearly departs from this behavior, with undersampling and oversampling of a factor of 10 and 6 at $S_M \pm 10°$, respectively. B08 has also strong undersampling below $S_M$ but has a landslide distribution that rapidly converges to the topographic one at high slopes. "

 in 3.1.3:  "We have observed that almost all of our eight events behave similarly with respect to the distribution of topographic slopes, not suggesting strong link with the individual storms properties. The C99 events has a different behavior and was indeed the shortest storm with the smallest total, but it was also the only cases occurring in high elevation terrain, with sparse vegetation. C15, the second shortest and smallest storm event may also have strong oversampling about 20° beyond $S_M$ but the limited number of landslides does not allow to confirm the significance of this oversampling."

We also updated accordingly the last paragraph of the discussion:

This framework might explain the preferential location on steep slopes observed for the very short duration C99 and possibly C15 (Fig \ref{slope}). However, the statistics of C15 are weak and C99 strong oversampling may relate mainly to specific mass movement triggered by surface runoff such as rilling and firehose (cf., Godt andt Coe, 2007). These processes also require high intensity, short duration events, but also low surficial infiltration rate leading to overland flow able to mobilize relatively loose surface materials.

"For other events, we analyzed the slope-gradient vs drainage area relationship for topography and landslide subset and did not find clear over-sampling of high-drainage and gentle gradient areas in the landslide distribution. It is well possible that a 30m DEM is not able to resolve accurately the fine-scale pattern of slope and drainage on the hillsides, where landslides occur, but it may also suggest that the upslope drainage area is not the main explanation. For example, the subsurface drainage efficiency may also increase with slope gradient, thus making very steep areas less likely to develop large pore pressure and possibly explaining the preferential landsliding of slopes just above the modal slopes for almost all events, indepedent of rainfall properties. Hydro-mechanical modeling at the catchment scale [e.g.,](von Ruette et al., 2013), applied on several of our dataset may be the only way to test between these different hypothesis."

==P17L9: T==otal storm rainfall: These results are extremely interesting. They suggest that absolute rainfall properties are good predictors for landslide properties. In the rainfall threshold literature there has been debate over whether absolute rainfall properties are driving failure or whether it is the degree of deviation from normal conditions (e.g. expressed as percentiles). It might be useful if you could reflect on this in relation to your findings. Would a plot of rainfall percentiles for these storms look very similar to the plot of absolutes that we see here?
>> **Rainfall percentiles are too difficult to establish for our database and within the scope of this review. However we added discussion on the relevance of normalized rainfall index, showed that storm total normalized by mean monthly rainfall correlates relatively well with landsliding.**
**We also underline in the mean text that although total rainfall may be a good predictor of the relative amount of landsliding between different storm ( as shown in Fig 6 and 7) the control on landsliding must be more compliacted as in the surrounding area similar rainfall occur without triggering landslides (Taiwan, Japan), or in the same**

**season similar total rainfall did not trigger landslide (in Colorado), so either antecedent rainfall or some constraints on intensity will be needed to generalize /strengthen the results we found.**

**We added in the main text : "However, we can envision that landscapes may rapidly reach an equilibrium in which all slopes unstable under rainfall conditions frequently occurring would have been removed. In this framework, the rainfall amount relative to the local climate would be more relevant than absolute rainfall, requiring an analysis in terms of deviation from the mean rainfall or in terms of rainfall percentiles (e.g., Guzzetti 2007). Although we could not define rainfall percentiles in each area, we note that normalizing $R_t$ by the mean monthly rainfall relevant for each storm, we still find a decent correlation with the peak landslide density, implying climate normalized rainfall variable may be driving landsliding (Fig Suppl. Z). We also note that even if $R_t$ allows to distinguish the magnitude of different landslide events additional constraints may be needeed to distinguish landslide from non-landslide event, given that for several cases we observe little or no landsliding in surrounding areas or during previous storms with similar rainfall amount. "**

P18L17: we have no clear physical explanation: isn't this something that either extreme rainfall community or the hurricane community have thought about? It would be useful to point readers to key reference from that literature here even if you don't strongly back one particular explanation.
**>> We replaced the original sentence ("We have no clear physical explanation of why this could be the case, and cannot exclude that it is a coincidence allowed by our small number of events.") by the following discussion :**

**"For tropical storms and hurricanes (5 out 9 cases in Fig 6D) a number of studies (cf., Jiang, et al., 2008 and ref therein), found that the maximum onland storm total rainfall (i.e, $R_t$ for us) correlated well (R>0.7) with a rainfall potential defined as the product of storm diameter, storm mean rainfall rate within this diameter over storm velocity, each terms measured 1-3 days before landfall. It was also generally observed that rainfall intensity is higher closer of the storm core, thus potentially tightening the link between $R_t$ and a given storm radius with intense rainfall and high landslide probability. These observations would imply linear proportionality between $R_t$ and $A_d$ and could be consistent with the observed power-law trend (1.5) (Fig 6), especially if some further links between $R_t$ and mean storm intensity or velocity exists. Potential links between $R_t$ and $A_d$ for smaller scale storms (C99,C15, B08 and B11) are harder to interpret, we cannot exclude that it is a coincidence allowed by our small number of events."**

**Jiang, H., Halverson, J. B., Simpson, J. and Zipser, E. J.: Hurricane "Rainfall Potential" Derived from Satellite Observations Aids Overland Rainfall Prediction, J. Appl. Meteor. Climatol., 47(4), 944–959, doi:10.1175/2007JAMC1619.1, 2008.**

Typographic errors and wording suggestions
P1
L10: storm -> storms
ok
L15: rainfalls -> rainfall
ok

P2
L7: At this day -> at present
Removed

L22: allow to make -> allow

ok

L23: region -> regions
ok
L23: (large slope -> (hillslope

L26: These progresses have - > this progress has#
ok
P3
L7: delete: , and thus
ok

L15: inventory > inventories
ok
L17: it isn't clear what you mean here, perhaps add: size (total area), geometry (length, width and depth) etc.
ok
L25 were > was
ok

P4
L15: N-s should be Ns
ok
L34: avoid > avoids
ok

P5
L2: twice more: or twice the number (i.e. 3n or 2n)?
>> corrected to "twice the number" (2n).
L31: you use a variety of date formats which is a little confusing.

L34: dates are needed for the FORMOSAT-2 image acquisition.
**>> They are given following MM/DD format. All are in 2009 as we specify now.**

P6: letters missing from Fig 1. Colours of landslides are very difficult to distinguish.
**>> Ok We are using BW hillshade to enhance landslide visibility.**
P7
L23-27: I don't think this is relevant here, I suggest moving to the discussion.
**>> We removed it. This already discussed somewhat in P17L24-26**
L30: average record properties: it isn't immediately clear what you mean here.
**>> We mean we take the average of the 3 gage records. Removing the word record may make it clearer: In each case we took the three closest gauges within 5 to 15km from the areas with the highest landslide densities (in 0.05 by 0.05° window) (Fig Suppl. 2) and computed their average properties**

L33: closest of > closest to
ok

P8
Table 1 caption: Reference are as follow > References are as follows
ok
L10: other > over
ok
L20: is > are
ok
P9
L1: polygons > polygon
ok
L3: allows > allows us

ok
P10
L25: the built > the
ok
L26: 0.2 and > 0.2 to
ok
P11
L10: with important total precipitation: this doesn't seem the right set of words
>> **With large total precipitation**
P13
L13: artifact due to > artefacts of
ok

P16
L4: S??: figure details missing.
Ok

P18
L23: prime > primary
ok
P21L5: storm tends > storms tend
ok

Anonymous Referee #2

General Comments

In this manuscript, the authors Marc et al. seek to understand what governs the spatial
and geometric characteristics of rainfall-induced landslides that result from single
storm events. Toward that end, they compile a database of landslide inventories from
single storm events spanning the past twenty years. The database is well considered
and thorough, and the strengths and limitations of each are discussed (e.g., availability
of local rain gauge data, etc.). With this dataset, the authors then compare landslide
spatial characteristics (number of landslides, area affected by landslides, landslide
density) and geometric characteristics (landslide total area, landslide scar area)
with precipitation characteristics (rainfall duration, storm intensity, total rainfall). In this
analysis, the authors make a number of interesting findings. For example, they find
that the longer-duration storms result in an increasing number of lower-gradient landslides.
Additionally, they show that while landslide volume and spatial density vary as
a nonlinear function of storm total rainfall, other landslide parameters do not appear
to depend on storm characteristics such as rainfall intensity. From this analysis the
authors conclude that their global inventory of single-event rainfall-induced landslides
can be queried to answer fundamental questions about the spatiotemporal evolution of
landslides in response to hydrologic forcing.
Understanding at a broad scale how a landscape may respond to a given storm event
is fundamentally useful for both geomorphic and hazard assessment applications, and
this topic should appeal to the readership of ESurf. The rigorous dataset compilation
by the authors provides a sound basis to start better quantifying these relationships,
and the statistical methods applied to the dataset are well founded and consistent with
those used by the community and should be readily reproducible by other researchers

(assuming the database will be available online). The observed correlations between storm magnitude and landslide area draw an interesting parallel to empirical studies of coseismic landslides (e.g., Keefer, 1994), and the data support the authors' conclusions both that rainfall magnitude (expressed as total storm rainfall) is a good indicator of landslide hazard potential and that the increase in lower-sloped landslides occurring over longer duration storms may reflect timescales for water to infiltrate lower portions of the landscape. Although the idea of rainfall-induced landslides occurring in lowersloping sections of a hillslope is well established (e.g., Reid and Iverson, 1992; Densmore and Hovius, 2000), this work shows that the prominence of this effect potentially depends on the storm magnitude and duration.

In terms of the general aspects of the manuscript presentation and layout, I find that the paper needs some fine-tuning and clarification, but overall it is close to being a finished product. The abstract provides a clear summary of the work, and the overall structure and segmentation of the manuscript is easy to follow. The title largely makes sense, although I find the last section stating "first insights from past and new events" confusing since all datasets are within the past twenty years and the youngest event occurred in 2015, and I'm not sure what that phrase adds to the description of the research. There are a number of places where language needs to be altered slightly, and I've tried to provide examples below in the technical comments section of the review. Mathematical formulae appear to be largely correct, but abbreviations for the landslide inventories (although intuitive) are not defined before they are used. Additionally, I found that Figure 2 should be modified, as it is very difficult to see the pink landslide polygons draped over the red and green topography. I imagine this would be especially difficult for people who are red-green colorblind. The supplementary material complements the manuscript well, and I have a few comments regarding supplementary figures below. Although I have a few additional concerns related to content and clarification, overall I think this paper will make an interesting contribution to Esurf..

**>> We thank the reviewer for its interest in our study and findings.**

**From these general comments we retain :**
**1/ A clarification of Figure 1 (not 2) showing the landslide inventories with a black and white hillshaded**

**2/ Some edits of the title : Following The discussion with Referee 1 we propose the following new title:**
**" Initial insights from a global database of rainfall-induced landslide inventories: the weak influence of slope and strong influence of total storm rainfall"**

**3/ Check that all landslide variables are defined in the text, and improve and correct texts, following both Referees technical comments.**

1st Paragraph: I'm not sure I agree with the statement that the goal of constraining quantitative relationships between landslide occurrence and rainfall is out of reach. The authors cite examples of this in the same paragraph. I do agree that there is certainly room for improvement in his area, which I think is the implied sentiment here.
**>> Out of reach was maybe exaggerated. We replaced "out of reach" by "difficult".**

Equation 1. I appreciate that the authors' goal here is to try to bridge the gap between purely deterministic models and purely statistical models, but I think that there needs to be a little more clarity. At the end of Paragraph 1, for example, you state that certain parameters such as permeability and cohesion that are required for deterministic approaches make a landscape-scale approach in data-poor regions inapplicable, yet you specifically include those parameters in your idealized semi-deterministic Equation 1. Why then is a deterministic approach not appropriate? I think that a bit more discussion might clarify these discrepancies.
**>> Deterministic approach will require a fine scale representation of porosity and its variability at a fine scale: more or less the one of the landslides, so at 10-100m. In a semi-deterministic approach we may need only a constrain on the mean porosity (and perhaps some other aspect of its distribution like its variance or skewness) within a whole catchment or 10x10km catchment. Obtaining such information remains a challenge but may be more tractable, and may be correlated to other large scale observable (From hydrological behavior to soil maps ?).**

We added "Note that variables in such equation may be statistical description at the catchment or landscape scale (being a simple mean or other moments of the distribution), and thus may not describe the finescale variability required by mechanistic models."

Lines 31-34: I'm confused by this sentence. When specifically are data from 2010-
2012 used? When May-June 2009 data are not available for a specific location in the
landslide-affected area?
**>> Exactly. We will clarify this, but in most places in this case imagery just after the event is available in Google Earth.**
**We rephrased : "... imagery available in Google Earth acquired in May-June 2009 in most areas, and in 2010-2012 elsewhere, where scars were still visible"**

Section 2.2 overall.
I'm also confused with the general methodology here. You map landslides on 30 m
Landsat imagery, as well as on higher resolution imagery within Google Earth, but
only in areas where a negative change in NDVI was observed at the 30 m scale. You
then say that field mapping in the area reports twice as many landslides than was
observed via remote sensing, but that the missing landslides must be smaller than #1
m resolution. Could the missing landslides not just be in areas that didn't result in a
negative NDVI shift in the landsat imagery? For example, a small translation or slump
in a forested area may not affect a 30 m pixel.
**>> This is true, it would however be relatively small landslide: indeed landslide much smaller than a landsat pixel ( e-g, ~5x5m rather than 30x30m) that looked fresh were almost systematically causing reduction of NDVI in one or 2 pixel, although the NDVI reduction was smaller than for large landslides.**
**So the Landsat NDVI is very sensitive to sub-pixel size landsliding, but the Google Earth imagery is essential to only map landslides and not many other anthropogenic/biological processes changing the NDVI.**

**We rephrased: "To avoid mapping post-event landslides, we mapped only the ones corresponding to vegetation radiometric index (e.g. NDVI) reduction for the pair of Landsat 5 images, present even for sub-pixel landslides (e.g., 10x5m)."**

Lines 23-27: In your discussion of peculiar landslide frequency distributions, you focus
on deviations from the (perhaps) expected Inverse-Gamma distributions at the large
end of the distributions. What about deviations on the smaller end? For example, in
the Total Area distributions, TW9, B11, C99, and J11 deviation from the maximum likelihood
estimations pretty substantially for small landslide areas. Converting total landslides
area to landslide scar area (As), the TW9 distribution especially deviates quite
far from the expected P values. Is there a known reason for these deviations? I am
far from an expert on landslide frequency distributions, but it seems worth discussing
since it is quite apparent on Figure 3!
**>> In agreement with Referee 1 we will also add a few line about the quality of IGD vs Lognormal. However, this will remain superficial given deciding on the functional form of size distribution is clearly out of the scope of our paper.**

**For the deviation in small landslide size, they are difficult to interpret because after the roll-over, censoring issues and difficulty to distinguish multiple adjacent landslides are likely important even with high resolution imagery.**

**No changes made.**

Figure 3: Similarly, I don't believe it is mentioned why the authors choose to break up
their landslide populations into two groups. Is this just to more easily visualize? Or is it
based on the quality of datasets?
**>> Indeed the split is just for visualization. We will specify it in the caption:**

**"To improve visualization we split the 8 inventories in two groups. "**

Lines 11-12: Is there a plot that shows the relationship described on these lines? I couldn't find one. Maybe it would be worth including these in the supplemental material.
**>> Ok we will add such a figure in the supplement.**

Lines 8-9: Very cool.
**>> Thank you, this is indeed exciting.**

Lines 12-15: If there is a continuous forcing of heavy rainfall over an extended period of time, it is not clear to me why a monsoon would not fit in with the scaling relationships derived in this paper. Would that not be an end-member condition for considering the role of water infiltration in setting the spatial distribution of landslides on lower slopes? If not, then why not? I imagine other people not as familiar with monsoon dynamics like myself might ask the same question.
**>> The reviewer is certainly right about the slope distribution, but not for the scaling between total rainfall and landsliding. To clarify the point we added:**
**Page 18 L20: "Indeed, in a long period with fluctuating rainfall such as the monsoon, drainage and storage of water will certainly not be negligible and the derivation of a soil water content proxy will be necessary (cf., Gabet et al., 2004) ."**

**Given the scaling between storm duration and slope sampling is not as clear as before after the revision of the method we do not comment further on it.**

Line 14: Does the proportion of flat ground affect the slide aspect ratios as well, since the flat ground may provide more accommodation space for runout?
**>> Locally maybe, but here these sentence really relate to the availability of topography within the footprint of the storm event dominated by the amount of submerged area or flat plains not by the width of valleys that may affect landslide runout.**

**No changes made.**

Line 22: This sentence cannot be true, as Figs. 6 and 7b all show a relationship between storm metrics on landslide scar areas. Do you mean other storm metrics outside of storm total rainfall?
**>> Here, the reviewer confuses the total area (counting only scar or whole landslide) shown in Fig 6A, 6B and 7, and the individual scar area distribution shown in Figure 3.**
**To avoid such confusion for other readers we rephrase to :**
**"We do not find a clear influence of storm metrics on the probability distribution of individual landslide scar areas or landslide runout (Fig 3)."**

Line 20: This is almost certainly true, especially for the smaller-area landslides that depend on local slope smaller than what a 30 m pixel can resolve.
**>> We also think that. Thus an analysis with a high resolution DEM may be needed.**

Technical Corrections

Line 3: "..we have very few datasets of rainfall-induced landslides." I think this should be clarified that this is the case only for single-event inventories.
**>> rephrased to : mostly because we have very few inventories of rainfall-induced landslides caused by single storms.**

Line 6: should be "orders of magnitude"
**>>ok**
Line 8: "The non-linear scaling with total rainfall." Two notes: 1) "nonlinear" is one word;
2) the variable that is being scaled with total rainfall should be specified.
**>>The nonlinear scaling of landslide density with total rainfall**
Line 11: "contrarily" should be "contrary"
**>>ok**
Line 18: "..itself expected to increase with global change." I find this sentence slightly
confusing. Consider replacing "itself" with "which are"
**>>Ok**
Line 26: should be "This progress has been possible"
**>>Ok**
Line 5: Should be "is needed"
**>>Ok**
Line 7: Sentence fragment ": : :affected by the storm, and thus . "
**>> This sentence has been replaced following Ref 1 : "Although relatively rare, some case studies based on
fragmentary event inentories exist (and are briefly reviewed in the next section) but they may contain too few
landslides for statistical analysis or may be biased to specific locations (along roads or near settlements, within weak
lithological units, downslope, etc), thus complicating the deconvolution of forcing and site influences."**

Lines 13-18: Nice overview!
**>>Ok**
Line 20: should be "datasets" (plural)
**>>Ok**
Line 25: should be "The rainfall was"
**>>Ok**
Line 30: "details" should be singular
**>>Ok**
Sections 2.1 and 2.2 – Introduce acronyms (e.g., B08) as you introduce each dataset.
**>>Ok**

Line 29: "(30m)" I think a space usually goes between the value and the units, e.g., "30
m". This should be done consistently throughout the manuscript.
Figure 1. It would be very helpful to make the landslide inventory polygons and rain
gauge stations contrast more with the background. The topography could easily be
represented as a hillshade since the absolute elevations are not the focus of the figure.
Also, the panels are not labeled with letters as described in the caption.
In the caption, "Landslides inventory" should be "Landslide inventories"
**>> We have homogenized space for units. We use now black and white hillshaded DEM for Fig 1 and included the
letters to label the panels.**

Line 2: "sub-parts" could be "subsets" perhaps?
**>>ok**

Lines 2-3: The values and units appear italicized in one instance and standard font in
the other. Should be consistently reported.
**>> All units will be in standard font.**

Lines 5-6: I think "rain gauge" is the correct spelling
**>>Ok**

Lines 13-14: Is the correlation between D and Rt and I3 and Rt shown anywhere? May
be good for the Supplemental.

**>> We added a supplementary figure for that.**

Line 21: Probably good to put a reference here where the increased runout for larger
volume slides is discussed (e.g., Legros, 2002).
**>> Here our point is not on the fact that large volume may have more runout, but rather that landslide with long runout may not obey the empirical relations V ~ aA^g (Larsen et al., 2010).**

Line 8: "However, for subset with less" should be "However, for subsets with fewer"
**>>Ok**
Line 10: "uncertainties" should be "uncertainty"
**>>Ok**

Line 1: Sentence could be simplified here, e.g., "Landslide inventories typically exhibit
heavy-tailed, power law frequency size distributions"
**>>This was modified following the other referee suggestion:  Frequency size distribution of landslide inventories have typically been fit by power-law tailed distributions,...**

Line 14: "Fitted" should be "Fit"
**>>Ok**

Line 9: the term "a slope gradient units" is confusing.
>> Rephrased to : **This ratio represents the tendency of landslide occurrence on a given slope to be more or less frequent than the expected occurrence of this given slope in the landscape.**

Line 10: before the word "oversampling", "and" should be "an"
**>>Ok**
Line 30: "create" should be "creates"
**>>Ok**
Line 34: Should there be a figure reference here?
**>>Ok Fig 6.**

Line 7: Should reference Fig. 5d here.
**>>Ok**

Figure 5: In Figure 5b, you could define the axis value more clearly on the axis itself.
"Landslide sampling on steep slopes" does paint a clear image of what the axis value
represents.
>> Figure 5B has been removed.

Unit labels are italicized and inconsistent with other unit labels throughout the text.

Line 4: remove "S??"
**>>Refers to Fig Suppl 4.  Corrected.**
Line 3: Could reference Fig 5b specifically.
**>>Ok**

[revised manuscript text omitted]

---

## Author Response (AR2)

Dear Editor,

 Please find a final, corrected version of the manuscript.

Following  the suggestions of the associate editors, I have read through the manuscript to implement basic grammar corrections and I have also tried to reduce the amount of details when discussing previous work: editing sentences in the introduction (on P2 L20 , P3 L16) and on section 2.1 (P4 L10-15).

I also added a sentence in the conclusion (P23 L30), reminding about climate normalization (Following on Referee 1 comments and the inclusion of Suppl Figure 8), that remain  an important open question.

Below is a Marked changed version of the manuscript.

I think the manuscript is ready to go for typesetting and proofs,

Sincerely,
Odin Marc on behalf of the other co-authors,

[revised manuscript text omitted]